

# Metrics for Intercomparison of Remapping Algorithms (MIRA) applied to Earth System Models

Vijay S. Mahadevan[1], Jorge E. Guerra[2], Xiangmin Jiao[3], Paul Kuberry[4], Yipeng Li[3], Paul Ullrich[5], Robert Jacob[1], Pavel Bochev[4], and Philip Jones[6]

[1]Mathematics and Computational Science Division, Argonne National Laboratory, Lemont, IL 60439, USA
[2]OU/CIMMS, NOAA National Severe Storms Laboratory, Norman, OK, USA
[3]Department of Applied Mathematics & Statistics, Stony Brook University, Stony Brook, NY 11704, USA
[4]Center for Computing Research, Sandia National Laboratories, P.O. Box 5800, Albuquerque, NM 87125, USA
[5]Department of Land, Air and Water Resources, University of California, Davis, CA 95616, USA
[6]Fluid Dynamics and Solid Mechanics Group, Los Alamos National Laboratory, Los Alamos, NM 87545, USA

**Correspondence:** V. S. Mahadevan (mahadevan@anl.gov)

**Abstract.** Strongly coupled nonlinear phenomena such as those described by Earth System Models (ESM) are composed of multiple component models with independent mesh topologies and scalable numerical solvers. A common operation in ESM is to remap or interpolate results from one component's computational mesh to another, e.g., from the atmosphere to the ocean, during the temporal integration of the coupled system. Several remapping schemes are currently in use or available for ESM. However, a unified approach to compare the properties of these different schemes has not been attempted previously. We present a rigorous methodology for the evaluation and intercomparison of remapping methods through an independently implemented suite of metrics that measure the ability of a method to adhere to constraints such as grid independence, monotonicity, global conservation, and local extrema or feature preservation. A comprehensive set of numerical evaluations are conducted based on a progression of scalar fields from idealized and smooth to more general climate data with strong discontinuities and strict bounds. We examine four remapping algorithms with distinct design approaches, namely ESMF Regrid (Hill et al., 2004), TempestRemap (Ullrich and Taylor, 2015), Generalized Moving-Least-Squares (GMLS) (Trask and Kuberry, 2020) with post-processing filters, and WLS-ENOR (Li et al., 2020). By repeated iterative application of the high-order remapping methods to the test fields, we verify the accuracy of each scheme in terms of their observed convergence order for smooth data and determine the bounded error propagation using the challenging, realistic field data on both uniform and regionally refined mesh cases. In addition to retaining high-order accuracy under idealized conditions, the methods also demonstrate robust remapping performance when dealing with non-smooth data. There is a failure to maintain monotonicity in the traditional $L^2$-minimization approaches used in ESMF and TempestRemap, in contrast to stable recovery through nonlinear filters used in both meshless GMLS, and hybrid mesh-based WLS-ENOR schemes. Local feature preservation analysis indicates that high-order methods perform better than low-order dissipative schemes for all test cases. The behavior of these remappers remains consistent when applied on regionally refined meshes, indicating mesh invariant implementations. The MIRA intercomparison protocol proposed in this paper and the detailed comparison of the four algorithms demonstrate that the new schemes, namely GMLS and WLS-ENOR, are competitive compared to standard conservative minimization methods requiring computation of





mesh intersections. The work presented in this paper provides a foundation that can be extended to include complex field definitions, realistic mesh topologies, and spectral element discretizations thereby allowing for a more complete analysis of production-ready remapping packages.

*Copyright statement.* The submitted manuscript has been created by UChicago Argonne, LLC, Operator of Argonne National Laboratory ("Argonne") and Sandia National Laboratories. Argonne, a U.S. Department of Energy Office of Science laboratory, is operated under
Contract No. DE-AC02-06CH11357, and Sandia National Laboratories is a multi-mission laboratory managed and operated by National Technology and Engineering Solutions of Sandia, LLC., a wholly owned subsidiary of Honeywell International, Inc., for the U.S. Department of Energy's National Nuclear Security Administration under contract DE-NA0003525.

The U.S. Government retains for itself, and others acting on its behalf, a paid-up nonexclusive, irrevocable worldwide license in said
article to reproduce, prepare derivative works, distribute copies to the public, and perform publicly and display publicly, by or on behalf of the Government. The U.S. Department of Energy (DOE) will provide public access to these results of federally sponsored research in accordance with the DOE Public Access Plan. http://energy.gov/downloads/doe-public-access-plan.

## 1   Introduction

Coupled multimodel simulations often involve high degrees of computationally complex workflows, and achieving consistently
accurate solutions is strongly dependent on the choice of spatiotemporal numerical algorithms used to resolve the interacting scales in physical models. Rigorous spatial coupling between components in such systems involves field transformations and communication of data across multiresolution grids while preserving key attributes of interest such as global integrals and local features, which is usually referred to as the process of *remapping* (also "regridding" or just "interpolation") (Van Leer, 1979; Dukowicz and Kodis, 1987; Jones, 1999). Such remap procedures are critical in ensuring the stability and accuracy of
scientific codes simulating multiphysics problems that typically occur in many different scientific domains. While there have been many high-order, stable interpolators proposed within different disciplines (Zienkiewicz and Zhu, 1992; Grandy, 1999; Jones, 1999; Smith et al., 2000; Garimella et al., 2007; de Boer et al., 2008; Slattery, 2016), to our knowledge, an effort to uniformly compare and measure the properties of these schemes has not yet been attempted. The current work is motivated by a need for an intercomparison of remapping schemes, which led us to standardize several numerical metrics to uniformly
compare the properties of these algorithms that are routinely applied to problems in climate and weather system modeling.

Remapping algorithms, in general, compute the spatial interpolation or quasi-interpolation of field data that is defined on a source mesh ($\Omega_s$) onto a target ($\Omega_t$) component mesh. It is imperative to note that these mesh pairs, $\Omega_s$ and $\Omega_t$, are generally of a different tessellation topology, spatial resolution, and regularity. Over the past several decades, there have been considerable efforts to create new conservative remapping algorithms to tackle various grid combinations and create consistently
accurate schemes for solution transfer between component models in multiphysics problems in many fields besides climate. These efforts have resulted in several software libraries and packages developed for this purpose. Examples include schemes developed for fluid-structure interaction (FSI) or heat transfer (such as MpCCI (Joppich and Kürschner, 2006) and preCICE



(Bungartz et al., 2016)), moving mesh problems with arbitrary Lagrangian-Eulerian (ALE) methods (Dukowicz and Kodis, 1987; Dukowicz and Baumgardner, 2000), and general-purpose remap software such as MOAB (Tautges and Caceres, 2009; Mahadevan et al., 2020), PANG (Gander and Japhet, 2013), and Data Transfer Kit (DTK) (Slattery et al., 2013).

Remapping packages developed for ESMs such as the Community Earth System Model (CESM) (Hurrell et al., 2013) and the Energy Exascale Earth System Model (E3SM) (E3SM Project, 2018), include SCRIP (Jones, 1999), the Earth System Modeling Framework (ESMF) Regridder (Hill et al., 2004), TempestRemap (Ullrich and Taylor, 2015; Ullrich et al., 2016), ncremap (Zender, 2008), Common Remapping Software (CoR) (Liu et al., 2018), and the YAC interpolator (Hanke et al., 2016) libraries. These remapping packages generate files with mapping weights that are then consumed by couplers such as MCT (Larson et al., 2005), cpl7 (Craig et al., 2012) and OASIS3-MCT (Craig et al., 2017), which apply the weights during runtime. Support for generation of mapping weights, and application at runtime is growing (Mahadevan et al., 2020).

Many of the production-ready remapping software implementations used in global climate simulations are typically based on first and second-order conservative mesh-based schemes, with additional support for second-order non-conservative bilinear patch reconstructions (Zienkiewicz and Zhu, 1992; Rider, 2014), or third-order bi-cubic spline interpolations (Hanke et al., 2016) for scalar fields. The projection algorithms implemented in these libraries for the conservative maps are based on the computation of an overlay or a "supermesh" (Jiao and Heath, 2004a; Farrell et al., 2009; Farrell and Maddison, 2011), defined as $\Omega_{S\cap T} = \Omega_s \cap \Omega_t$, which is then used to compute consistent and conservative linear operators through $L^2$-minimization for transferring field information (Ullrich and Taylor, 2015; Ullrich et al., 2016).

Remapping also occurs in other parts of a climate model, such as within the geophysical fluid dynamics solver of a component. Remapping strategies for tracer transport advection schemes such as the Semi-Lagrangian Inherently Conserving and Efficient (SLICE) scheme (Zerroukat et al., 2005), Conservative and Monotone Cascade Remap on Sphere for CubedSphere (CS) to regular Latitude-Longitude (RLL) grids (Lauritzen and Nair, 2008), geometrically exact conservative remapping approaches (Ullrich et al., 2009), and some other high-order mass conserving schemes (Norman and Nair, 2008; Erath et al., 2013) have also been devised. While some of the previous work, including conservative semi-Lagrangian multi-tracer transport schemes (CSLAM) (Lauritzen et al., 2010) with second-order variants (Zerroukat et al., 2006), can be easily extended to arbitrary mesh topology combinations, many of the other approaches are tied to specific cases and not used in the inter-component coupling, which falls out of the scope of this study.

While traditional mesh-based schemes requiring computation of overlay/exchange grids have the advantage of being inherently conservative (Balaji et al., 2006), they tend to be computationally expensive (Jansen et al., 1992) and achieving high-order accuracy for scalar field data can sometimes pose difficulties. Numerically suboptimal approximations, such as nodal expansions or nearest neighbor values, can be used to compute interpolators quite efficiently if the field being remapped is smooth and does not mandate conservation prescriptions. Bilinear and bicubic interpolations can also prove to be alternatives here. However, consistently ensuring high-order accuracy under conservation constraints is often necessary for many real-world fields, such as the global heat flux that is exchanged between the atmosphere and ocean components in climate systems.

More recently, new mesh-based (Li et al., 2020) and fully meshless (Trask and Kuberry, 2020) remapping schemes that do not require overlap meshes have been developed for other problems in order to mitigate the computational complexity of





computing exact intersections between unstructured meshes. These algorithms can provide high-order, conservative and stable alternatives for climate modeling compared to the traditional linear maps generated from $L^2$-minimization approaches.

As the number of available remapping algorithms grows, it becomes imperative to compare them under a unified framework to understand the properties of the schemes, before applying them to real-world simulations. Additionally, while the compu-
tational cost is critically important for production runs, our intercomparison study specifically compares only the numerical performance of each algorithm under varying mesh topologies and field regularity, which are closely representative of those from a climate model such as E3SM. The presented protocol hence provides a systematic way to test and compare all existing and new remapping algorithms being developed for earth system modeling.

**Organization**

The current study aims to better understand and document the key properties of the remapping schemes through the use of mesh, field, and scale resolution-independent metrics definitions. This intercomparison paper is organized as follows:

– A detailed literature survey on the current state-of-art remapping schemes used in climate modeling and various re-lated coupled physics problems, along with the relevant numerical background for four specific high-order remapping algorithms and their implementations, is presented in Section 2.

– Definitions for remapping metrics that evaluate field accuracy, global conservation, strict global bounds control, and feature dispersion are presented in Section 3. We argue that these metrics are broadly useful for evaluating key properties that determine the accuracy and stability of the solution transfers between model components, and can be applied to all remapping algorithms widely used in climate codes.

– A sample workflow for computing and comparing various numerical metrics in a unified and unbiased fashion is featured
in Section 4. Some implementation details on the open-source Python-based infrastructure are also provided.

– Consolidated results from the intercomparison study applied to four competing remap algorithms on representative problems are shown in Section 5 for both uniformly refined and regionally refined global meshes.

– Potential future research directions to extend the analysis presented in this work are described in Section 6, followed by a summary of the intercomparison study in Section 7.

**2   Remapping Algorithms**

In general, for coupled simulations, we need to transfer a field $\psi_s \in \mathbf{R}^{n_s}$ defined on the source grid, $\Omega_s$, to a quantity $\psi_t \in \mathbf{R}^{n_t}$ on the target grid, $\Omega_t$. A remap is hence defined as an operator $R : \mathbf{R}^{n_s} \mapsto \mathbf{R}^{n_t}$ that transfers $\psi_s$ to a *trial* field $\widetilde{\psi}_t \in \mathbf{R}^{n_t}$. As noted above, we generally wish to preserve a *property* defined as an operator $P : \mathbf{R}^{n_t} \to \mathbf{R}^m$, that describes discrete physical invariants and/or inequalities relevant to $\psi_t$. However, to keep the current work focused we restrict attention to definitions that
are applicable only to scalar fields.





Finally, the accuracy or order of the remap typically depends on how the field $\psi$ is reconstructed from a set of discrete values. On a mesh, this could be a Taylor series expansion around a cell centroid (Jones, 1999) or a general Finite-Element (FE) type expansion using nodal basis functions.

## 2.1  Related Work

One of the simplest data-transfer methods is to use piecewise interpolation functionals. This approach is particularly convenient if $\psi_s$ defined on $\Omega_s$ has an associated function space, which can be directly utilized for evaluating the interpolator. Such consistent interpolation techniques utilizing the underlying basis functions for field descriptions give rise to standard second-order linear (on simplicial elements) or bilinear (with quadrilateral elements) interpolations (Hill et al., 2004). In finite-volume methods, or for interpolating from station data ("scattered-data" or "Location Streams"), the nearest-point interpolation is

sometimes used, which is at best first-order accurate.

More generally, the remap methods designed for scattered data and cell-to-cell transfers are closely related to the *reconstruction* problem, i.e., to reconstruct a piecewise smooth function or its approximations at some discrete points on a mesh, given some known quantities at discrete locations on the same mesh. The fundamental difference between reconstruction and remapping is that the former involves a mapping between a discrete and a continuous function space, while the latter involves

a mapping between two discrete spaces defined on $\Omega_s$, and $\Omega_t$. The demonstrated techniques used in high-order interpolators use spline quasi-interpolants (de Boor, 1990), bi-cubic splines (Hanke et al., 2016; Craig et al., 2017), the standard radial-basis function spaces (Flyer and Wright, 2007; Bungartz et al., 2016), the moving least-squares (MLS) method (Lancaster and Salkauskas, 1981), and its variants such as the modified MLS (MMLS) (Joldes et al., 2015; Slattery, 2016), which originate from high-order reconstruction methods. Recent extensions to MLS for producing efficient high-order remap involve reconstructing

locally the manifold geometry from a point set representation, and then generating a compact stencil in the local coordinate chart (Liang and Zhao, 2013; Suchde and Kuhnert, 2019; Trask and Kuberry, 2020; Gross et al., 2020).

More recently, Li et al. (2020) proposed a high-order remap technique for piecewise smooth functions on surfaces, known as *WLS-ENO remap* (WLS-ENOR), which is based on the *continuous moving frame* (*CMF*) for smooth functions (Jiao and Wang, 2012) as well as an ENO-style technique (Liu and Jiao, 2016) for resolving discontinuities. For smooth functions, CMF

differs from MLS (Lancaster and Salkauskas, 1981) in that it uses compact stencils over local coordinate systems from a $\mathcal{C}^0$ normal field, instead of global stencils and $\mathcal{C}^\infty$ moving frames.

Typically, a remap method can and should be independent of the discretization methods used in physics models. In this context, the method may be nonconservative, or conservative with respect to certain properties of the reconstruction. A major disadvantage of the *quasi-interpolation* techniques is the lack of inherent constraints for conserving energy during the remap

process; see, e.g., de Boer et al. (2008); Jiao and Heath (2004a). Note that the high-order quasi-interpolants (with degree of basis expansion $p > 1$) tend to be significantly less dissipative than low-order methods ($p \in [0, 1]$); see, e.g., Slattery (2016). However, such remap methods are not guaranteed to produce conservative or monotone remapped solution fields.

A common approach to overcome this issue is to enforce *conservation*, so that the integrals over the source and target meshes are equal. There are several examples of first- and second-order conservative remap schemes (Chesshire and Henshaw,





1994; Grandy, 1999; Gander and Japhet, 2013; Jones, 1999; Ullrich and Taylor, 2015) that rely on common-refinement-based $L^2$ projection (Jiao and Heath, 2004a) approaches. These schemes require computation of a function integral defined on the source mesh over some control volumes associated with a target node (or cell). The numerical integration is computed over the intersections of the elements (or cells) of the source and target meshes. These intersections form the common refinement

(Jiao and Heath, 2004a), or supermesh (Farrell et al., 2009), whose computations require sophisticated computational-geometry algorithms for efficiency and robustness (Jiao and Heath, 2004b, c). Although these first- and second-order schemes applied to ESMs are conservative, they may exhibit excessive numerical diffusion resulting in dissipation of "energy", especially near field discontinuities, or regions with large second derivatives.

    Concurrently, when applying high-order remapping methods, discontinuities in the function defined on $\Omega_s$ can lead to over-

shoots or undershoots when evaluated on $\Omega_t$. The discontinuities may be in the function values (aka $\mathcal{C}^0$ discontinuities), which tend to lead to $\mathcal{O}(1)$ oscillations (or "ringing") that do not vanish under mesh refinement, analogous to the Gibbs phenomena (Gottlieb and Shu, 1997; Fornberg and Flyer, 2007). Additionally, any discontinuities in the derivatives ($\mathcal{C}^1$ discontinuities) tend to cause milder oscillations that vanish under mesh refinement but nevertheless may accumulate in repeated remapping cycles. It is often critical to resolve or control these numerical artifacts as they introduce non-physical variations that can

influence the numerical stability of the coupled global nonlinear multiphysics system.

    The deficiency in linear mapping approaches arises from the fact that they are only dependent on $\Omega_s$, and $\Omega_t$, but completely independent of the solution field that is being projected between the meshes. As a consequence of Godunov's theorem (Godunov, 1959) extended to linear remapping workflows with monotonicity constraints, there may be a restriction on the optimal achievable order of accuracy as shown by Van Leer (1979) while still preserving global solution bounds during the projection

step. Hence, often the properties of the linear maps can be enhanced by using a procedure that is nonlinear (depending on the projected field variations) to recover property preservation for high gradient fields.

    In this vein, the techniques for resolving Gibbs phenomena can be classified as *filtering* and *mollification*; see Jerri (2013). The filtering techniques often rely on post-processing, such as cropping and property redistribution, to ensure local conservation in a neighborhood. A recent variation of the mass borrowing approach (Royer, 1986) uses a limiter as a filter during the

remapping process (Bradley et al., 2019), in order to impose local bounds preservation for linear map applications, such that monotonicity can be recovered even when the underlying remapper does not provide it. This "Clip-And-Assured-Sum" (CAAS) post-processing filter can be useful to avoid spurious numerical oscillations due to resolution disparity, or strong gradients in the underlying solution. Applying CAAS filters on quasi-interpolatory linear maps can hence produce strictly bounded reconstructions on $\Omega_t$, while providing property preservation, as a viable option to achieve better remapped solutions

in production climate simulations. In contrast, mollification adapts the kernels (i.e., basis functions) near discontinuities. In the past, a discontinuity detecting, *a posteriori* stabilization procedure has been used with specific mesh discretizations (Blanchard and Loubere, 2016) to choose optimal orders of reconstruction adaptively in order to ensure better behavior for polygonal meshes. Other methods, such as WLS-ENOR, detect discontinuous regions and can effectively adapt the weighting schemes to resolve the Gibbs phenomena at the cost of additional computations at runtime.





Alternatively, rather than imposing a weakly nonlinear post-processing filter, using fully nonlinear remap schemes can be an option when the computational cost of the solution transfer is not the dominant factor in the simulation. Such nonlinear remap schemes typically use optimization-based remap (OBR) procedures (Carey et al., 2001; Bochev and Shashkov, 2005; Bochev et al., 2011) to minimize the net residual projection error using Lagrange multipliers. OBR follows a "divide-and-conquer"

alternative (Bochev et al., 2014) to direct property preserving methods, which separates accuracy considerations from property preservation. In so doing, OBR helps to avoid interdependencies between mesh quality, accuracy, and property preservation that force many monotone reconstruction methods to make trade-offs between the latter two (Berger et al., 2005). Extensions of such schemes for remapping climate data is an unexplored topic and may provide avenues for future research.

Additionally, mimetic schemes that use compatible function spaces (Thuburn et al., 2009) depending on the fields being

transferred between component models have proven to be extremely accurate for remapping scalar and vector fields on Arakawa C/D grids (Pletzer and Hayek, 2019). Potential extensions for arbitrary mesh topologies to yield compatible, conservative remaps for fluxes in climate components are possible (Ringler et al., 2010). However, to our knowledge, general theory and implementations for remapping on arbitrary meshes are currently unavailable, which may restrict the usage of such schemes for production climate simulations.

## 2.2   Weighted Least-Squares Approximations in Remapping

A common theme across all of the remapping methods described in this paper is that they utilize some variants of the least squares approximations (aka quasi-interpolation) in their computational kernels. To illustrate the idea, let us consider a function $f(\boldsymbol{u}) : \Omega \to \mathbb{R}$ at a given point $\boldsymbol{u}_0 = [0,0]^T$, such as a quadrature point in a cell on the target mesh. Let us first assume a domain $\Omega \subset \mathbb{R}^2$ for simplicity, where $\boldsymbol{u} \equiv [u,v]$. Let $f$ be $\mathcal{C}^{p-1}$ continuous for some degree $p \geq 0$, then $f(\boldsymbol{u})$ can be approximated to

$p+1^{st}$ order accuracy about $\boldsymbol{u}_0$, using a degree-$p$ two-dimensional Taylor polynomial as

$$f(\boldsymbol{u}) = \sum_{q=0}^{p} \sum_{\substack{j+k=q \\ j,k \geq 0}} c_{jk} u^j v^k + \mathcal{O}\left(h^{p+1}\right), \tag{1}$$

where $c_{jk} = \dfrac{1}{j!k!} \dfrac{\partial^{j+k}}{\partial u^j \partial v^k} f(\boldsymbol{u}_0)$, and $h$ is a measure of the radius of the local neighborhood. In cell-to-cell transfer, typically some integrals of $f$ are known on the source mesh. Given $m$ cells on the source mesh in a neighborhood of a point $\boldsymbol{u}_0$ on the target mesh, and let $\phi_i(\boldsymbol{u})$ be the test function (such as the Heaviside functions) associated with the $i$th cell. Let $b_i$ denote these

known integral values. We then obtain a system of $m$ equations with $n = (p+1)(p+2)/2$ unknown coefficients $c_{jk}$,

$$\int_{\Omega} \sum_{q=0}^{p} \sum_{\substack{j+k=q \\ j,k \geq 0}} c_{jk} u_i^j v_i^k \phi_i(\boldsymbol{u}) \, \mathrm{d}\boldsymbol{u} \approx b_i \tag{2}$$

for $1 \leq i \leq m$. Note that one can also use a bi-degree-$p$ Taylor polynomial by letting $0 \leq j \leq p$ and $0 \leq k \leq p$ in (1), which would lead to $n = (p+1)(p+1)$ unknowns. The resulting approximate Taylor polynomial can then be used, for example, to evaluate (or quasi-interpolate) $f$ at $u_0$ to $p+1^{st}$ order accuracy, or even to higher order due to superconvergence. The





same procedure can also be applied to construct a trivariate quasi-interpolation by replacing $\boldsymbol{u}$ and $u^j v^k$ with $\boldsymbol{x}$ and $x^j y^k z^\ell$, respectively. The $m$ equations in (2) can be written in matrix form as $\boldsymbol{Ax} \approx \boldsymbol{b}$, where $\boldsymbol{A} \in \mathbb{R}^{m \times n}$ is known as a *generalized Vandermonde matrix*, $\boldsymbol{x} \in \mathbb{R}^n$ is composed of $c_{jk}$ in (1), and $\boldsymbol{b} \in \mathbb{R}^m$ is composed of the known integrals $b_i$ on the source mesh. In general, the generalized Vandermonde system (2) is rectangular. It can be solved by minimizing a weighted norm of

the residual vector $\boldsymbol{r} = \boldsymbol{b} - \boldsymbol{Ax}$, i.e.,

$$\min_{\boldsymbol{x}} \|\boldsymbol{r}\|_{\boldsymbol{W}} \equiv \min_{\boldsymbol{x}} \|\boldsymbol{W}(\boldsymbol{Ax} - \boldsymbol{b})\|_2, \tag{3}$$

where $\boldsymbol{W}$ is a diagonal matrix containing the weight for each row in $\boldsymbol{A}$. This formulation is known as *weighted least squares* or *WLS* in short (Golub and Van Loan, 2013). When $m = n$, and $\boldsymbol{A}$ and $\boldsymbol{W}$ are nonsingular, $\boldsymbol{W}$ does not affect the solution. More generally, when $m \neq n$, different $\boldsymbol{W}$ leads to different solutions by changing the relative importance of certain sample points.

Different remapping algorithms and reconstruction schemes use specific weighting strategies to achieve optimal solutions to the weighted least-squares problem in Equation (3). Note that the generalized Vandermonde matrix $\boldsymbol{A}$ may be ill-conditioned as the polynomial degree $p$ used for the reconstruction grows. A preferred approach to address this potential ill-conditioning is to use a rank-revealing QR factorization (Golub and Van Loan, 2013; Li et al., 2020).

### 2.3    Algorithms for Earth System Models

Among the standard conservative remapping and high-order reconstruction strategies introduced in Section 2.1 for climate problems, we selected four specific algorithms to explore in detail. The motivation for choosing these algorithmic implementations was driven by their high usability including the use in current ESM's, and the rigorous underlying numerics that can be verified and validated consistently for a large suite of test problems. We categorize these algorithms below into three broad groups based on whether the algorithms require overlay meshes and whether mesh data structures are utilized to compute the

solution reconstruction.

I) *Overlay-mesh-based remappers*

We consider two specific implementations that provide conservative remapping capability for production ESMs.

(a) ESMF: The Earth System Modeling Framework's *Regrid* function providing bilinear and conservative maps

(b) TempestRemap: Conservative, consistent, and monotone remapper with higher-order $L^2$ projection support

Both ESMF Regrid and TempestRemap provide the remapping capability for scalar fields defined on $\Omega_s$ based on a weighted residual formulation given in Equation (3). With a source field $\psi_s$ defined on $\Omega_s$, the formulation computes $\psi_t$ on $\Omega_t$ by solving the following problem:

$$\int_\Omega \psi_t \phi_i \, \mathrm{d}V = \int_\Omega \psi_s \phi_i \, \mathrm{d}V, \tag{4}$$





where the $\phi_i$ are suitable weight functions. In a common-refinement-based transfer (Jiao and Heath, 2004a), the $\phi_i$ are the basis functions $\psi_{t,i}$ on $\Omega_t$, which leads to a *Galerkin projection method*. Such a Galerkin method can achieve conservation globally by solving the weighted residual minimization in Equation (4). More details on the specific methodologies used for reconstruction in these implementations are provided in later sections.

Note that the SCRIP library (Jones, 1999) also does provide both first- and second-order conservative remapping schemes, but it was not included in the current study since its capabilities are similar to that of ESMF and TempestRemap implementations, and those two are used in current production versions of CESM and E3SM.

II) *Meshless remappers*

Reconstruction methods that do not directly utilize the topological information about the underlying mesh layout are
*meshless* methods. In our study, we will consider the Generalized Moving Least-Squares method (GMLS), with global conservation, monotonicity constraints, and local bounds preservation provided by CAAS as a post-processing filter.

Future studies could also include the comparison of MMLS (Slattery, 2016) and RBF (Bungartz et al., 2016) reconstructions within this framework for remapping climate data, since those methods have demonstrated some success in fluid-structure interaction and nuclear engineering problems.

III) *Non-overlay mesh-based hybrid remappers*

The final category includes the mesh-based remappers that do not require computation of an intersection mesh between $\Omega_s$ and $\Omega_t$. The Weighted Least-Squares Essentially Non-Oscillatory Remap method (WLS-ENOR) utilizes a discontinuity capturing, high-order technique for piecewise smooth functions to produce optimal field transformations between meshes by minimizing the residual in Equation (3).

Other opportunities for comparison in this category include using reconstructed climate data with the conservative bilinear algorithm in ESMF or bicubic interpolations available in YAC (Hanke et al., 2016).

These algorithms and their implementations span a range of remapping techniques, including those used currently in production runs and those that can potentially be used in the future given the availability of open-source software. Further details regarding the numerics and the software tools for each of the schemes are provided in the following subsections.

## 2.3.1   ESMF

The Earth System Modeling Framework (ESMF[*]) (Hill et al., 2004) consists of a suite of software modeling tools that support building Earth System Models. Among other features, ESMF exposes several key functionalities to transfer data between component models in weather and climate modeling. It offers a variety of data structures for transferring field data between components, and libraries for regridding, time advancement, and other common modeling functions. The advanced regridding
algorithms provided by ESMF implement standard bilinear interpolation, higher-order patch recovery, first and second-order

---

[*]ESMF: https://earthsystemmodeling.org/





conservative projections, and several variations of non-conservative nearest neighbor interpolants. ESMF can produce maps that are "offline", in the sense that can be precomputed, stored, and then applied as a linear operators to arbitrary fields defined on $\Omega_s$ to compute the projection on $\Omega_t$. Fully online remapping is also possible with ESMF.

In the current intercomparison study, we utilized the command line applications installed with ESMF (version 8.1) to gen-
erate and apply interpolation weights from the command line using NetCDF files. Using this ESMF_REGRIDWEIGHTGEN application, interpolation weights can be generated for various grid combinations and applied to source field data to compute projections between component models. More specifically, we select only the first and second-order conservative projection methods invoked with 'conserve' and 'conserve2nd' command-line options respectively, which are implemented in ESMF for scalar fields and fluxes. These options are routinely used in production climate models and hence provide valuable baselines on
the current state of remapping algorithms in our comparison study. For more details regarding the numerics and implementation of the algorithms in ESMF, we refer readers to Collins et al. (2005), Jones (1999), and Kritsikis et al. (2017).

### 2.3.2 TempestRemap

TempestRemap[†] is a software library to generate conservative, consistent, and monotone remapping weights of arbitrarily high order accuracy between meshes on the sphere. Similar to the ESMF tools, TempestRemap generates "offline" maps that
are consumed by ESMs. TempestRemap supports the computation of remapping weights between any combination of finite volume and/or spectral element (both continuous and discontinuous spaces) descriptions with conservative prescriptions. For purposes of the current manuscript, we describe only the high-order FV-FV algorithms below.

In TempestRemap, the procedure used to generate remapping weights for FV discretizations consists of two primary operations (Ullrich and Taylor, 2015; Ullrich et al., 2016). First, local polynomial reconstructions are defined over the source mesh
with some adjustments made for spherical geometry (Jalali and Gooch, 2013). The coefficients of these local reconstructions are computed according to a weighted pseudoinverse method (Skamarock and Menchaca, 2010; Skamarock and Gassmann, 2011). These polynomials are integrated over the target mesh by using the overlap, or supermesh (Farrell et al., 2009), which in general can be defined as $\Omega_{s\cap t} = \Omega_s \cap \Omega_t$. Note that when computing high-order linear maps, if a sufficiently large patch on the source map is not used during the reconstruction process, the condition number of the generalized Vandermonde matrix
$A$ in Equation (2) can be very high, leading to numerical roundoff errors and degradation in the overall accuracy of the remap operator to first-order accuracy in the neighborhood.

A common approach for the integration operator has been to transform integrals over mesh faces into line integrals over the boundary using the divergence theorem (Dukowicz and Kodis, 1987). While this technique has been used to define conservative remapping operators, a drawback is that it requires an analytical expression for the potential function, which can be difficult in
general. An alternative method was proposed in Erath et al. (2013), which, although avoiding some of the difficulties of the line integral approach, lacks consistency. In TempestRemap, a quadrature-based integration operator is used to generate an initial set of remapping coefficients which is guaranteed to be consistent (exactly remaps the unit field), but may not be conservative.

---

[†]TempestRemap: https://github.com/ClimateGlobalChange/tempestremap





To obtain conservation, the coefficients of the matrix are projected linearly into the space of maps that are both conservative and consistent.

### 2.3.3 GMLS

The Generalized Moving Least Squares (GMLS) method extends the Moving Least Squares (MLS) technique from approxi-
mation of point values to approximation of arbitrary linear functionals (Nayroles et al., 1992; Breitkopf et al., 2000; Wendland, 2004; Mirzaei et al., 2012). Similar to MLS, a distance-based weighting kernel is used to favor the source data sites closer to the point of query or reconstruction. GMLS features a broad choice of sampling functionals and target operators, and hence the term 'generalized' in its name. The application of nonlinear sampling functionals and target operators for GMLS is possible (Gross et al., 2020), but is certainly not common. In fact, most applications of GMLS use linear sampling functionals and
target operators (gradient, curl, divergence, integral average along an edge or over a cell, etc...), for which there is theory on approximation and wellposedness. When the sampling functionals and the target operator are simply pointwise evaluations, GMLS reduces to the traditional MLS method.

High-order accurate approximations cannot be achieved with traditional MLS schemes using function spaces described by Raviart-Thomas ($\mathcal{H}(div)$), and Nedelec basis ($\mathcal{H}(curl)$), or even cell-averaged quantities that are common in FV codes. How-
ever, through careful selection of the sampling functionals and target operators, consistent approximations of fields embedded in these various non-standard spaces are possible using the GMLS approach. This flexibility of the GMLS method greatly increases the available types of input data that can be handled to produce high-order reconstructions of fields between $\Omega_s$, and $\Omega_t$.

When a GMLS problem is solved, a set of coefficients corresponding to elements of a basis (traditionally polynomial) are
computed. The target operation is then applied to each member of the basis used for approximation, after which a dot product is made between the originally computed coefficients and the evaluation of the target operation acting on the basis. While GMLS permits a wide choice of target operators, we choose a target operation which is the cell average of each member of the approximating basis on the target grid.

Traditionally, GMLS uses a basis that is defined as a function of the spatial dimension from which a point cloud is sampled.
However, in this work, reconstruction of functions sampled on a manifold permits generating a compact stencil in a local coordinate chart, which is one dimension smaller (Liang and Zhao, 2013; Suchde and Kuhnert, 2019; Trask and Kuberry, 2020; Gross et al., 2020). The savings in net computational floating-point operations (flops) is a factor of $p^3$ in $R^2$, as compared to a traditional basis in $R^3$, where $p$ is the order of the basis.

As is true for many other regression schemes, GMLS is not inherently conservative to machine precision, but rather it
is "conservative" to discretization precision. In other words, the degree to which it violates conservation is discretization-dependent and generally vanishes with refinement. However, such weak conservation notions may not be deemed satisfactory for climate modeling applications where exact global conservation has a history of being demanded and valued. In such cases, GMLS remap should be followed by a post-process filter to restore global conservation to machine precision. In this paper, we use either the GMLS or GMLS-CAAS notations to indicate whether the CAAS routine has been used as a post-processing filter





after each remap step, in order to restore global conservation, global bounds, along with an attempt to improve local property preservation.

Similar to the overlay-mesh-based, ESMF, and TempestRemap offline remappers described in Section 2.3.1 and Section 2.3.2 respectively, the solution of the GMLS problem produces a stencil or a linear mapping that can be computed *a priori* to a
simulation. This enables storage of the relevant parts of the remap operator as a sparse matrix. Therefore, the use of GMLS without post-process filtering can be thought of as an offline remapper, and GMLS-CAAS can combine the offline and online processes to achieve more favorable properties. Note that the CAAS filter is only one of several choices available for the online post-processing filter. Alternative strategies such as the nonlinear OBR (Bochev et al., 2014) for feature detection, and minimization of local dissipation may be possible in conjunction with the GMLS workflow. These enhanced GMLS remapping
strategies may be considered in future studies.

There are several key motivations for using GMLS to perform field remapping for ESMs. These include mesh topology independence, as well as flexibility in the choice of the sampling functional, and the target operator, thereby enabling remap of non-traditional degrees of freedom that may be defined on the vertices, edges, or faces of $\Omega_s$. Additionally, the embarrassingly parallel nature of the dense linear systems that require inversions during reconstruction yields a high flops-to-communication
ratio. This feature makes GMLS more suited for next-generation platforms. Given the computationally intensive nature of the GMLS algorithm, it is best implemented in libraries focused on parallel performance portability, computational efficiency, and significant compiler optimizations. The Compadre Toolkit[‡] (Kuberry et al., 2019) is built on Kokkos (Edwards et al., 2014) and Kokkos Kernels, such that a single version of the code is written to be performant on both multi-core CPU and hybrid-GPU architectures. Compadre is built to assemble and solve large batches of GMLS problems in parallel, thereby leveraging batched
QR solvers with pivoting for the parallel solution of many small dense linear systems.

### 2.3.4   WLS-ENOR

*WLS-ENOR*, or *Weighted-Least-Squares-based Essentially Non-Oscillatory Remap* (Li et al., 2020) is based on the WLS-ENO reconstruction technique proposed in (Liu and Jiao, 2016). Originally developed for solving hyperbolic PDEs, WLS-ENO detects discontinuities and then reduces or eliminates the potential Gibbs phenomena in the solutions of the PDEs by adapting
the weighting schemes in WLS based on the local features in the solution. WLS-ENOR adapted WLS-ENO to remap data between meshes, and in the process, it enhanced the treatment of discontinuities to resolve not only the severe oscillations at $\mathcal{C}^0$ discontinuities (i.e., jumps in function values), but also the accumulated effect of mild oscillations due to $\mathcal{C}^1$ discontinuities (i.e., jumps in the derivatives).

Unlike the preceding remapping techniques, WLS-ENOR is a non-overlay mesh-based technique in that it uses the mesh for
computing numerical integration, but it does not require an overlay mesh (although it has the option to use the overlay mesh if available). More specifically, WLS-ENOR utilizes adaptive quadrature rules with $p$-refinement in smooth regions and $h$-refinement near discontinuities in its numerical integration to achieve high-order accuracy and (near-local) conservation. More specifically, WLS-ENOR is composed of three major components. The first component is a WLS-based algorithm for smooth

---

[‡]Compadre Toolkit: https://github.com/SNLComputation/compadre





functions, as we described in Section 2.2. In this context, the weighting scheme in WLS-ENOR is based on a positive-definite radial function due to Buhmann (Buhmann, 2001). As shown in (Li et al., 2020), this weighting scheme encourages statistical error cancellations and enables better superconvergence (i.e., higher than $(p+1)$st-order convergence) with even-degree $p$ for node-to-node transfer of smooth functions. In this inter-comparison work, we use an extension of the work in (Li et al.,

2020) to cell-centered data. Mathematically, this extension essentially uses the step functions (aka Heaviside functions) over the cells on the source mesh as the test functions in (2), compared to the Dirac delta function at the nodes on the source mesh as test functions in (Li et al., 2020). Note that in this work, we apply WLS on a sphere $\Omega = S$, which is topologically a 2-D object embedded in $\mathbb{R}^3$. One could construct a WLS in $\mathbb{R}^3$ directly as in (Slattery et al., 2013), which would lead to a large generalized Vandermonde system. Alternatively, we can construct a local $uv$ (surface tangent) coordinate frame at each point

$\boldsymbol{x}_0 \in S$. Specifically, let $\boldsymbol{m}_0$ denote an approximate normal at $\boldsymbol{x}_0$, which can be the exact normal to $S$ or a first-order estimation. Let $\boldsymbol{t}_1$ and $\boldsymbol{t}_2$ form an orthonormal basis of an approximate tangent plane orthogonal to $\boldsymbol{m}_0$. The local $uv$ coordinate frame is then centered at $\boldsymbol{x}_0$ with axes $\boldsymbol{t}_1$ and $\boldsymbol{t}_2$. We can then transform the sample points about $\boldsymbol{x}_0$ to use this local $uv$ coordinate frame and apply the WLS to construct a bivariate quasi-interpolation. In terms of implementation, WLS-ENOR constructs a matrix-based transfer operator for smooth functions, which maps the cell-averaged values from the source mesh to the target

mesh.

The second component in WLS-ENOR is the detection and resolution of discontinuities. In particular, WLS-ENOR can detect $\mathcal{C}^0$ and $\mathcal{C}^1$ discontinuities of the function on the source mesh and then transfer the discontinuity tags from the source mesh to the target mesh. The detector in WLS-ENOR is based on an asymptotic analysis of the Gibbs phenomena near $\mathcal{C}^0$ and $\mathcal{C}^1$ discontinuities as described in (Li et al., 2020). The original detector in (Li et al., 2020), however, was for node-to-

node transfer. For cell-to-cell transfer in this work, we simply apply the detector on the dual mesh and treat cell-averaged values as approximations of cell-center values. This approximation is second-order accurate, which is sufficient in detecting discontinuities. After identifying the discontinuities, WLS-ENOR uses a solution-based weighting scheme that effectively leads to (nearly) one-sided quasi-interpolation in discontinuous regions. This solution-based weighting scheme causes WLS-ENOR to use a different set of basis functions to overcome the Gibbs phenomena, and hence it can be classified as a mollification

technique (cf. Section 2.1). In terms of implementation, the discontinuity detector on the source mesh is implemented as a matrix-based operator; after the discontinuities are identified, a new local solution-based transfer operator is constructed for each cell on the target mesh near a discontinuous cell on the source mesh. We refer readers to (Li et al., 2020) for details.

The third component in WLS-ENOR is an adaptive quadrature technique, which is enabled when the target mesh is significantly coarser than the source mesh. In this case, simply sampling the function at the quadrature points of a target cell may

miss some important local features on the source mesh, especially near discontinuities. To overcome this loss of information, one could use the overlay mesh as in TempestRemap. This approach may be ideal in terms of accuracy and conservation, but it introduces complications when the elements have curved edges. Although WLS-ENOR implementation supports this option, in this comparative work, we use a non-overlay-based version of WLS-ENOR that utilizes $h$-, and $p$-refinement of the quadrature rules in discontinuous and smooth regions, respectively. In particular, near the detected discontinuities, WLS-ENOR

subdivides the cells (i.e., $h$-refinement) on the target mesh to match the local resolution on the source mesh and then utilizes





the quadrature points of the sub-divided cells. For smooth regions, we found it more efficient to use the quadrature points of higher-degree quadrature rules (i.e., $p$-refinement) than subdividing the cells. This adaptive quadrature technique not only overcomes the loss of accuracy but also enables WLS-ENOR to recover global conservation in a fashion that is nearly local to discontinuous regions. To this end, if there is a gain or loss in terms of global conservation, we distribute this global error

proportionally to cells that have gained or lost mass locally, correspondingly. This conservation-recovery procedure reduces the pollution of the conservation errors from discontinuous regions to smooth regions. We estimate the local conservation errors using the subdivided cells near discontinuities for accuracy and robustness; for smooth regions, we use a simple comparison of the local extreme values in the local neighborhood for smooth regions for efficiency.

The current implementation of the WLS-ENOR algorithm uses MATLAB, where the core components were converted into

C using 'MATLAB Coder' (version 4.2). An open-source C++ implementation is currently underway and will be released in the future for both node-to-node and cell-to-cell field transfers.

## 3    Metrics to Evaluate Remapping Algorithms

Solution remapping on unstructured meshes is a complex process, and it is critical to satisfy several key properties to ensure that the transferred field data between components do not introduce unbounded and nonphysical error modes. In order to compare

different remapping schemes in an unbiased framework, we introduce five primary categories under which the comparison metrics can be grouped.

1. Sensitivity: Algorithmic invariance to underlying component mesh topology

2. Consistency: Ability to retain the order of convergence of the underlying discretization in a given norm

3. Conservation: Ensuring global integral, (and) local conservation of critical quantities

4. Monotonicity: Preservation of global and local solution bounds during remap and solution transfer between components

5. Dissipation: Minimization of local solution dispersion on repeated remap transfers between $\Omega_s$ and $\Omega_t$

Given a continuous field $\psi$, we use $\mathbf{D}^s$ and $\mathbf{D}^t$ to denote the reference discretization of $\psi$ on the source and target grid, respectively. These are generated by directly sampling the analytical fields and by Spherical Harmonic expansion (see section 4) for the realistic fields. The regridding operator from source to target mesh is denoted by $\mathbf{R}$, *i.e.* the regridded field $\psi$ is

denoted by $\mathbf{R}\mathbf{D}^s[\psi]$. The global integral operator is denoted by $I^s$ and $I^t$ on the source and target grid, respectively. Typically these operators take the form

$$I[\mathbf{x}] = \sum_{\text{all } j} x_j J_j, \tag{5}$$

where $J_j$ denotes the area or weight of the j$^{th}$ degree of freedom (i.e. the volume of a finite volume).



## 3.1 Grid Sensitivity

A crucial factor for the success and broader usability of a general remapping algorithm in ESMs is the ability to produce mesh-independent numerical behavior that is robust for any pair of structured or unstructured meshes. In other words, remapping algorithms need to be general and without approximations targeted at specific topological elements.

5    In the current work, we utilized three different meshes of varying resolutions. Specifically, we present analysis performed to compare remapping schemes using the Cubed-Sphere (CS), Quasi-uniform Voronoi (MPAS), Regular Latitude-Longitude (RLL) meshes, on both quasi-uniform and Regionally Refined Meshes (RRM). Some sample meshes used in the study are shown in Figure 1.

(a) Equiangular Cubed-Sphere mesh          (b) Quasi-uniform Centroidal Voronoi mesh          (c) Regular Latitude-Longitude mesh

(d) Cubed-Sphere RRM          (e) MPAS Voronoi RRM

**Figure 1.** A depiction of the five meshes studied in validation, unit testing and intercomparison of the regridding schemes.





## 3.2 Standard Accuracy Measures

Accuracy in the remapped solution will be assessed with standard error metrics defined as follows.

$$\|E\|_{L_1} \equiv \frac{I^t\left[\mathbf{e}_K\right]}{I^t\left[\left|\mathbf{D}^t[\psi]\right|\right]}, \tag{6}$$

$$\|E\|_{L_2} \equiv \sqrt{\frac{I^t\left[\mathbf{e}_K^2\right]}{I^t\left[\left|\mathbf{D}^t[\psi]\right|^2\right]}}, \tag{7}$$

$$\|E\|_{L_\infty} \equiv \frac{\max\left[\mathbf{e}_K\right]}{\max\left|\mathbf{D}^t[\psi]\right|}, \tag{8}$$

where $\mathbf{e}_K = \left|\mathbf{R}\mathbf{D}^s[\psi_K] - \mathbf{D}^t[\psi_K]\right|$ is the local remapping error in element $K \in \Omega_t$, $E$ represents the discrete error vector in the solution field relative to the reference field sampled on $\Omega_t$, and $I^t$ is the weighted integral using Equation (5) on $\Omega_t$. In some general sense, the error measure $\|E\|_{L_1}$ identifies errors in large-scale features, $\|E\|_{L_2}$ identifies errors in small-scale features, and $\|E\|_{L_\infty}$ identifies the largest pointwise error. Related to accuracy, *consistency* is assessed by applying these metrics to uniformly smooth fields with no $C^0$ or $C^1$ discontinuities, and verifying the asymptotic theoretical rate of convergence under uniform mesh refinement conditions.

Note that in order to eliminate potential aliasing errors, the normalization factors $\mathbf{D}^t[\psi]$ used in the denominator for definitions of $\|E\|_{L_1}$, $\|E\|_{L_2}$, and $\|E\|_{L_\infty}$ are computed based on the exact sampling (element-averaged for FV discretization) of the data (reference solution) on $\Omega_t$, and not using the projection of the field $R\mathbf{D}^s[\psi] \in \Omega_t$.

## 3.3 Gradient Preservation Measures

Preservation of the solution gradients in addition to other critical properties, such as local conservation in the remapping procedure, require $C^1$ continuity in the $D^s[\psi]$. Let $\nabla\mathbf{D}^s, \nabla\mathbf{R}\mathbf{D}^t$, be the gradients of the scalar fields on $\Omega_s$, and $\Omega_t$, respectively.

Then, in order to measure accuracy of the solution and its gradient, we introduce two specific global metrics: $|E|_{H_1}$ seminorm and the $\|E\|_{H_1}$ norm.

$$|E|_{H_1}^2 \equiv \frac{I^t\left[\nabla\mathbf{e}_K\nabla\mathbf{e}_K\right]}{I^t\left[\left|\mathbf{D}^t[\psi]\right|^2\right]}, \tag{9}$$

$$\|E\|_{H_1}^2 \equiv \frac{I^t\left[\mathbf{e}_K^2 + \nabla\mathbf{e}_K\nabla\mathbf{e}_K\right]}{I^t\left[\left|\mathbf{D}^t[\psi]\right|^2\right]} = \|E\|_{L_2}^2 + |E|_{H_1}^2, \tag{10}$$

where $\mathbf{e}_K = \left|\mathbf{R}\mathbf{D}^s[\psi_K] - \mathbf{D}^t[\psi_K]\right|$ is the local remapping error in element $K$, $\nabla\mathbf{e}_K = \left|\nabla\mathbf{R}\mathbf{D}^s[\psi_K] - \nabla\mathbf{D}^t[\psi_K]\right|$ is the corresponding gradient of the error, and $I^t$ is the weighted integral using Equation (5) on $\Omega_t$.





In this study, $\nabla q$ for some scalar $q$, defined as a cell mean value, is generated by finding the convex hull of cells surrounding the current cell and computing the gradient per Barth and Jespersen (1989). We compute these gradients as part of the metrics evaluation after each remapping sequence.

### 3.4 Global conservation

Global conservation is trivially assessed by evaluating the change in the global integral of the scalar field value on the source mesh and the projected field on the target mesh. We use the following metric to quantify global conservation.

$$L_g \equiv \frac{I^t\left[\mathbf{R}\mathbf{D}^s[\psi]\right] - I^s\left[\mathbf{D}^s[\psi]\right]}{I^s\left[|\mathbf{D}^s[\psi]|\right]}. \tag{11}$$

However, we note that this definition for $L_g$ is only meaningful when the target domain fully envelops the source domain (which may have gaps or holes in more general cases). In the case of climate modeling, an admissible example for using

Equation (11) would be for remapping heat and moisture fluxes from the land surface to the overlying atmosphere.

### 3.5 Global Extrema Preservation

Global extrema preservation can be assessed via the standard $G_{min}$ and $G_{max}$ error metrics (Ullrich and Taylor, 2015):

$$|G_{min}| \equiv \min\left\{0, \frac{\min\left(\mathbf{D}^t[\psi]\right) - \min\left(\mathbf{R}\mathbf{D}^s[\psi]\right)}{\max\left|\mathbf{D}^t[\psi]\right| - \min\left|\mathbf{D}^t[\psi]\right|}\right\}, \tag{12}$$

$$|G_{max}| \equiv \max\left\{0, \frac{\max\left(\mathbf{R}\mathbf{D}^s[\psi]\right) - \max\left(\mathbf{D}^t[\psi]\right)}{\max\left|\mathbf{D}^t[\psi]\right| - \min\left|\mathbf{D}^t[\psi]\right|}\right\} \tag{13}$$

The error measures $G_{min}$ and $G_{max}$ identify undershoots and overshoots, respectively, by taking on nonzero values when there is a departure away from the reference global extreme values. In other words, a nonzero value of the metric indicates changes in global extrema, indicating presence of Gibbs phenomenon. Hence, the global extrema metric is particularly useful as it provides indications about the monotonicity-preserving properties of the remapping schemes.

### 3.6 Local Extrema Preservation

Local extrema preservation can be assessed using a localized difference, *i.e.* to what degree does the remapped grid cell value fall within the range of surrounding grid cells sampled on the target grid. This consideration motivates us to define a localized difference in extrema:

$$\Delta_{min,j} \equiv \min\left\{0, (\mathbf{R}\mathbf{D}^s[\psi])_j - \min_{\text{1-ring patch}}\left(\mathbf{D}^t[\psi]\right)_j\right\}, \forall j \in \Omega_t, \tag{14}$$

$$\Delta_{max,j} \equiv \max\left\{0, (\mathbf{R}\mathbf{D}^s[\psi])_j - \max_{\text{1-ring patch}}\left(\mathbf{D}^t[\psi]\right)_j\right\}, \forall j \in \Omega_t. \tag{15}$$





where the minimum and maximum are taken over all grid cells on the target mesh that surround the current target cell $j$. These values can then be reduced to a single value in the usual manner by using an appropriate norm definition for both $\Delta_{min,j}$ and $\Delta_{max,j}$:

$$
\begin{aligned}
L_{min,1} &\equiv \frac{I^t[|\Delta_{min}|]}{I^t[|\mathbf{D}^t[\psi]|]}, & L_{max,1} &\equiv \frac{I^t[|\Delta_{max}|]}{I^t[|\mathbf{D}^t[\psi]|]}, \\
L_{min,2} &\equiv \frac{\sqrt{I^t[\Delta_{min}^2]}}{\sqrt{I^t\left[\left|\mathbf{D}^t[\psi]\right|^2\right]}}, & L_{max,2} &\equiv \frac{\sqrt{I^t[\Delta_{max}^2]}}{\sqrt{I^t\left[\left|\mathbf{D}^t[\psi]\right|^2\right]}}, \\
L_{min,\infty} &\equiv \frac{\max_j |\Delta_{min,j}|}{\max|\mathbf{D}^t[\psi]| - \min|\mathbf{D}^t[\psi]|}. & L_{max,\infty} &\equiv \frac{\max_j |\Delta_{max,j}|}{\max|\mathbf{D}^t[\psi]| - \min|\mathbf{D}^t[\psi]|}.
\end{aligned}
\tag{16}
$$

Note that the definition of the localized differences shown in Equation (14) and Equation (15) utilize a local neighborhood to determine the deviation from reference extrema values. This is sufficient to capture resolution of sharp gradients in the remapped fields under mesh refinement, for element-averaged data. However, the metric contains a $\mathcal{O}(h)$ dependence on the mesh resolution and can be applied to $\mathcal{C}^0$ or smoother fields, but not when $\mathcal{C}^0$ discontinuities are present.

## 4   Metrics for Intercomparison of Remapping Algorithms (MIRA) Workflow

For all remapping algorithms evaluated in this comparison study, we conduct iterative two-way ($\Omega_s \rightarrow \Omega_t$ and $\Omega_t \rightarrow \Omega_s$) remapping of an initial source field with FV discretization on $\Omega_s$. We do so in order to characterize the stability of a scheme and expose any dissipation effects, which would not be possible to ascertain when comparing single, uni-directional field transfers. With this workflow, we seek to quantify the consistency, stability, and convergence of each participating algorithm as measured with the metrics defined in Section 3. Here, $N_r$ indicates the number of iterative applications of the linear map to compute field transformation on $\Omega_s \rightarrow \Omega_t \rightarrow \Omega_s$, which will be referred to as *remap iterations*. The additional remapping metrics computed from the repeated remap iterations are critical in determining the long-term temporal behavior of a fully coupled climate simulation, as it entails multiple transfers of field data between component models. Hence, determining the stability of the remapping operator and understanding dissipation behavior for repeated transfers provide valuable insight into the propagation of spatial coupling errors in the solver.

### 4.1   Open-source MIRA Implementation

The workflow necessary to evaluate a given remapping method comprises of five consecutive steps described below.

    1. Generate a series of meshes of different topologies and resolutions. We use the Cubed-Sphere (CS), Quasi-uniform Voronoi (MPAS), Regular Latitude-Longitude (RLL), and Regionally Refined (RMM) grids of the CS and/or MPAS types of varying resolutions to devise the test cases. See figure 1 for an illustration of the meshes. The mesh data is stored in universal NetCDF4 format containing an array of vertex point locations and a cell connectivity map to describe the topology.





2. Given a collection of meshes as in step 1 above, a Python module called MESHPREPROCESSDRIVER is then used to generate and store the adjacency maps and unstructured cell area integrals with high-order Gauss quadrature rules. The convex hull map for each cell is also precomputed and stored during this step in order to speed up the evaluation of remapped field gradient metrics.

3. A second Python module called FIELDGENDRIVER then takes each of the pre-processed mesh files and evaluates scalar fields by sampling from either an analytical function on the sphere or a set of prescribed Spherical-Harmonic (SPH) coefficients, which is described in Section 4.2. In this step, a cell average is computed by local quadrature within each mesh element to a given order of accuracy by appropriately choosing the order of the quadrature to resolve the SPH expansion order. This operation is performed on all the input meshes to generate the reference "ground truth" realization of a given field, which is used to compute the metrics defined in Section 3 accurately.

   We emphasize that any existing mesh (such as Yin-Yang (Kageyama and Sato, 2004) or cubic-octahedral (Gaussian) reduced grids) can be used in this workflow, instead of the ones generated in step 1, as FIELDGENDRIVER only relies on existence of element connectivities and adjacency maps to be available for computing cell integrals.

4. All remapping algorithms evaluated in this study use the mesh data, depending on the scheme, and initial reference solutions on $\Omega_s$ to execute the test suite over one or many $N_r$ iterations. The expected output from each of the algorithms, for the test problems devised, are the discrete solution vectors $\psi_s^i \in \Omega_s$, and $\psi_t^i \in \Omega_t$, where $i \in [1, \ldots N_r]$. In the current study, unless otherwise specified, $N_r = 1000$.

5. The final Python module in the metrics suite, METRICSDRIVER, can then be invoked on each of the remapped output data (typically stored in NetCDF files) to compute all the remapping metrics defined in Section 3 consistently. The computed remapping metrics are then stored as comma-separated values (CSV) for further analysis and intercomparison studies.

The schematic shown in Figure 2 provides further details on this workflow. Note that in order to evaluate and compare a new remapping implementation such as SCRIP (Jones, 1999), CoR (Liu et al., 2018), or YAC (Hanke et al., 2016), only the fourth and fifth steps in the workflow have to be executed, since the pre-processed input meshes and the sampled reference data have been made available publicly (Mahadevan et al., 2021).

## 4.2 Scalar Test Variables on the Sphere

In the remapping intercomparison study, we consider five scalar test variables defined on the sphere as reference solutions fields. These fields are chosen such that different aspects of the remapper can be evaluated uniformly. Details about the analytical and real-world fields and the sampling methodology used in the Python implementation are provided below.





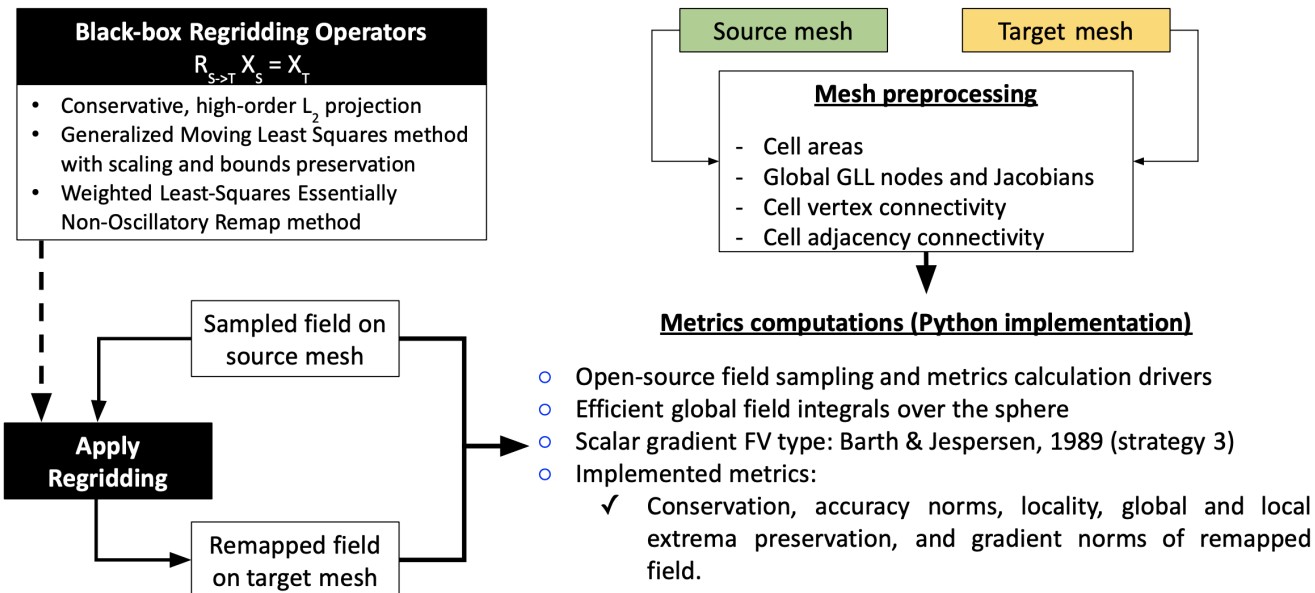

**Figure 2.** The MIRA workflow for generating the remapping metrics for the intercomparison study.

### 4.2.1 Idealized Analytical Fields

Idealized fields used in this study mirror the approach of Lauritzen et al. (2007) and Ullrich and Taylor (2015). Namely, we employ three idealized test cases of varying complexity to understand the error measures produced by remapping. The two analytical fields studied are depicted in Figure 3.

The first analytical field (ANALYTICALFUN1) is a combination of spherical harmonics functions with frequency wave similar to order 3, given by

$$\psi = Y_3^2 + Y_3^3, \tag{17}$$

where $Y_m^l$ are the real spherical harmonic functions evaluated through the SHTOOLS package for degree $m$ and polynomial order $l$.

Following Jones (1999), and Lauritzen et al., (2007), the second field (ANALYTICALFUN2) is a relatively smooth function resembling spherical harmonics of order 2, and azimuthal wavenumber 2, given by

$$\psi = 2 + \cos^2\theta \cos(2\lambda), \qquad (Y_2^2). \tag{18}$$

These fields are used to test performance for a smooth, well-resolved field and a slightly high-frequency, weakly-resolved field with rapidly changing gradients. Given that the analytical expressions for these fields are trivial to evaluate, we can

compute the exact numerical errors introduced by the remapping schemes when projecting the fields from $\Omega_s$ to $\Omega_t$.



Note that the FIELDGENDRIVER module can take arbitrary closed-form functions and evaluate them on the sphere by using high-order quadrature order rules to sample and compute element averaged data. This design allows the flexibility to test slightly more complex analytical vortex fields (Ullrich et al., 2009) or any three-dimensional real-valued function projected on the sphere with coordinate transformations (Townsend et al., 2016).

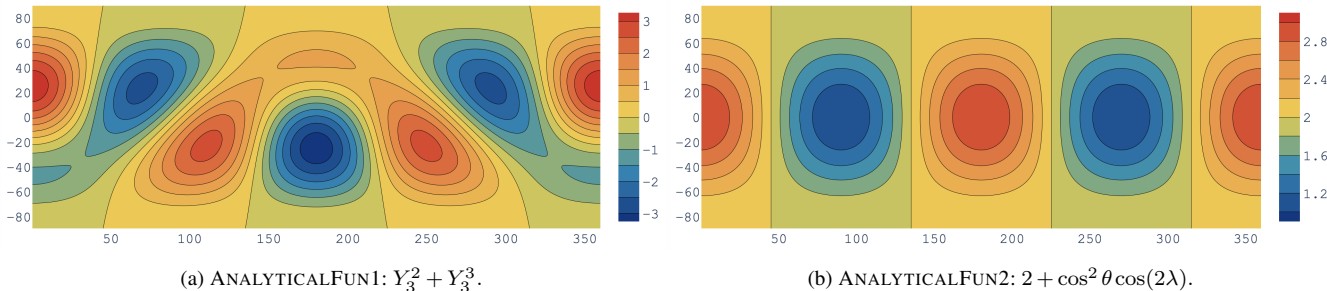

(a) ANALYTICALFUN1: $Y_3^2 + Y_3^3$.          (b) ANALYTICALFUN2: $2 + \cos^2 \theta \cos(2\lambda)$.

**Figure 3.** Contour plots of analytical fields used in this study; ANALYTICALFUN1 (left), and ANALYTICALFUN1 (right)

### 4.2.2 Real Data Fields

We also test the performance of each remap technique by regridding real data fields obtained from freely available composite satellite observations. The fields chosen are: Total Precipitable Water (TPW), Cloud Fraction (CFR), and Global Topography (TOPO). From a representative data set, we compute a Spherical Harmonic (SPH) decomposition in order to determine an analytical approximation of spectral content. We employ these particular fields because we can control their characteristics as functions, i.e., global bounds for topography, positive definiteness for precipitable water, and continuity for cloud fraction. As such, these fields present distinct challenges to the remapping methods. For convenience, we use the SHTOOLS (Wieczorech and Meschede, 2018) package through its Python interface (PySHTools), which facilitates the computation of spectra based on spherical harmonic bases and reconstruction of fields thereof.

Given the 1D averaged spatial amplitude spectrum for each set of composite satellite data as shown in Figure 4 with a corresponding linear fit, we then produce controlled randomized realizations for each field on any unstructured mesh and resolution, including regionally refined grids. The randomization is applied to the coefficients of the expansion at each degree, extracted from the linear fit functions in Figure 4, and is entirely reproducible given an integer seed to a pseudo-random number generator (set to 384 in our study). The code for this is available in the FIELDGENDRIVER (Guerra et al., 2021). The original data ranges from 0.25 to 0.1 degrees of resolution and SPH reconstructions are smoothly varying up to the number of coefficients employed. Note that while the order of SPH reconstruction can be specified arbitrarily high (between 1 and 512 modes) to get a better resolution of the fields, the computation of element-averaged sampling representative of FV discretization needs to utilize a sufficiently high-degree quadrature rule such that the SPH expansions are exactly integrated on the sphere.



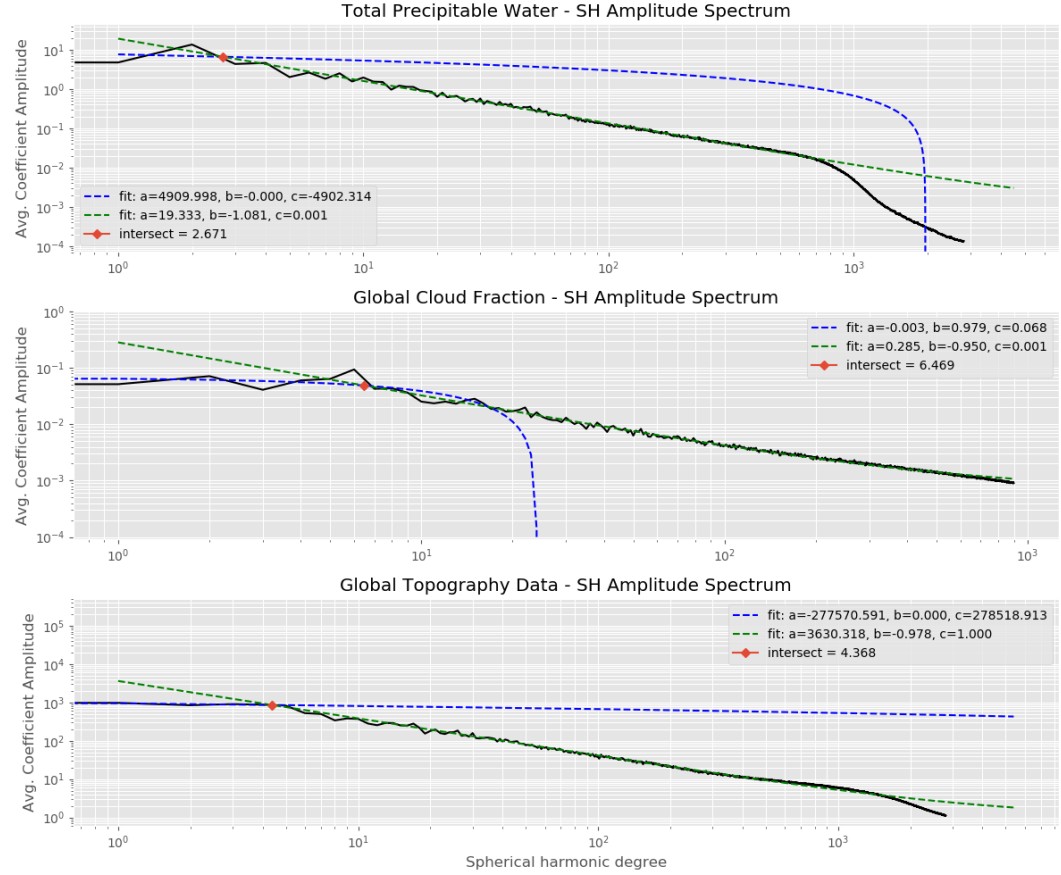

**Figure 4.** 1D averaged spatial amplitude spectrum for real data fields based on composite satellite observations. Fit coefficients over two branches (dotted blue and green lines) correspond to a function: $f = ax^b + c$ where $x$ is the logarithm of the spherical harmonic degree.

**Total Precipitable Water (TPW):** Global composite data for TPW is taken from the MIMIC-TPW2 project[§] (Wimmers and Velden, 2011). Morphological compositing is applied to microwave sensor data and numerically advected. This field has an absolute minimum for all realizations of 0.0 mm with maxima typically in the 70.0 to 80.0 mm range. A representative, randomized reconstruction of TPW is shown in Figure 5a.

5  **Cloud Fraction (CFR):** Global composite data for CFR is taken from the NASA AQUA/MODIS data archive[¶] (Platnick et al., 2020). This field uses absolute global limits for all realizations of 0.0 to 1.0. Thus, by imposing these bounds after each reconstruction, $\mathcal{C}^0$ discontinuities are introduced. These manifest as spatially flat regions in the data where maximum cloud cover is noted. A randomized reconstruction of CFR is shown in Figure 5b.

---

[§]MIMIC-TPW2 project: http://tropic.ssec.wisc.edu/real-time/mtpw2/product.php?color_type=tpw_nrl_colors&prod=global2×pan=24hrs
[¶]NASA AQUA/MODIS data archive: https://neo.sci.gsfc.nasa.gov/view.php?datasetId=MYDAL2_M_CLD_FR





(a) Global TPW test field for remapping metrics

(b) Global CFR test field for remapping metrics.

(c) Global TOPO test field for remapping metrics.

**Figure 5.** Randomized reconstruction of the real-world fields used in this study.

**Global Topography (TOPO):** Global topography data is taken from the ETOPO1 Global Relief Model[‖] (Center, 2009; Amante and Eakins, 2009). This field has a global minimum/maximum for all realizations of -10994.0/8848.0 m, but is otherwise smoothly varying. A randomized reconstruction of TOPO is shown in Figure 5c.

The raw satellite data snapshot of TPW, CFR, and TOPO fields that are used in the current study have been made available

5   separately (Guerra and Mahadevan, 2021) in order to make the workflow reproducible.

---

[‖]ETOPO1 Global Relief Model: https://www.ngdc.noaa.gov/mgg/global/global.html





## 5 Results and Discussions

Comparing different remapping algorithms under a unified infrastructure for test problems and metrics collection is a nontrivial task. The metrics defined in this study and the implementation of the various field samplings on arbitrary unstructured meshes have provided large output datasets to analyze the key properties of the remappers under consideration.

Specifically, a series of different mesh types (CS, MPAS, RLL) $N_{type}^{uni} = 3$ with five different refinement levels were generated to cover both the uniform refinement cases ($N_{ref}^{uni} = 5$) and regionally refined cases with CS-MPAS mesh type ($N_{type}^{rrm} = 2, N_{ref}^{rrm} = 3$) around the continental US. Using the field definitions ($N_{fields} = 5$) introduced in Section 4.2, consisting of two smooth analytical fields and three real-fields expanded with spherical harmonics, sampling was then evaluated on all input meshes ($N_{type}^{uni} N_{ref}^{uni} + N_{type}^{rrm} N_{ref}^{rrm} = 15 + 6 = 21$) to serve as reference solutions. With these input meshes (Mahadevan et al.,

2021), the remapping algorithms were applied following the workflow in Figure 2 for combinations of CS-MPAS (both uniform and RRM), MPAS-RLL, and RLL-CS meshes of varying resolutions to evaluate the metrics data.

    The volume of consolidated output metrics data is enormous from this experiment, since 1000 remapping iterations were performed on $N = N_{type}^{uni} [N_{ref}^{uni}]^2 N_{fields} = 375$ global, uniformly refined mesh cases, and $N = [N_{ref}^{rrm}]^2 N_{fields} = 45$ RRM cases, with each of the four remapping algorithms using various degrees of reconstructions $p$ for FV-FV field transfers. Mea-

suring more than 15 different remapping metrics for each of these cases has provided extensively detailed results to compare the algorithmic implementations in an unbiased fashion. Hence, unless explicitly noted, only significantly unique results are presented and discussed below in the following subsections. We direct readers to the IPython notebooks available in (Mahadevan et al., 2021), which can be used to generate the comparison for any combination of mesh types, source resolution, target resolution, field variable, and remapping schemes. Note that all the metrics data collected during the analysis are also stored in

the same repository for reproducibility.

    Detailed results from the intercomparison study and discussion on the implication of each metric to the remapping scheme are presented next.

### 5.1 Consistency

The consistency of the high-order remap algorithm implementations can be verified by remapping smooth functions and cal-

culating the spatial convergence order of the resultant approximations on the target mesh after repeated remaps. Using the sampled analytical functions described by ANALYTICALFUN1 and ANALYTICALFUN2, a verification study was conducted using the standard error norms and gradient error metrics data for the various schemes. For smooth solution profiles, the theoretically expected convergence rates for a consistent remapping method are $\mathcal{O}(h^{p+1})$ in $\|E\|_{L_1}$, $\|E\|_{L_2}$, and $\|E\|_{L_\infty}$ global error norms, and $\mathcal{O}(h^p)$ in $\|E\|_{H_1}$ and $|E|_{H_1}$ global gradient error norms.

In these studies, we observed that convergence rates for both low and high-degree approximations up to $p = 2$ of the analytically smooth fields show good agreement with the theoretically expected accuracy convergence rates. However, for high-order mesh-based remaps, achieving higher than $\mathcal{O}(h^3)$ convergence rate appears to be a limitation with the methods involved in this





study, while the hybrid and meshless schemes show consistent recovery of the approximation orders for all degrees tested. The following subsections provide detailed results and discussions for each of the remapping algorithms.

### 5.1.1 ESMF

The conservative schemes implemented in the ESMF package (Hill et al., 2004) have been thoroughly verified, and are routinely used to generate the linear maps for solution transfer between components in E3SM. The second-order conservative projection algorithm that was originally introduced for ALE computations (Dukowicz and Kodis, 1987), and later applied to spherical meshes (Jones, 1999), has been implemented in ESMF with an appropriate linear gradient reconstruction (Kritsikis et al., 2017). We measure the convergence rates for both the first-order ('conserve') and second-order ('conserve2nd') conservative schemes and present the results observed in Table 1. The ESMF first-order conservative scheme yields expected rates. However, the second-order scheme shows degraded convergence rates as confirmed by the global error and gradient norms. This convergence result is unexpected and contrary to the analysis of the second-order, piece-wise linear finite-volume reconstruction procedure presented by (Kritsikis et al., 2017), which is implemented in ESMF.

**Table 1.** ESMF: Convergence rates for ANALYTICALFUN1 field on all mesh types

| ESMF Option | Grid Type | $\|E\|_{L_1}$ | $\|E\|_{L_2}$ | $\|E\|_{L_\infty}$ | $\|E\|_{H_1}$ | $|E|_{H_1}$ |
|---|---|---|---|---|---|---|
| **conserve** | CS-MPAS | 1.031 | 1.022 | 0.857 | 0.004 | 0.002 |
| | MPAS-RLL | 1.034 | 1.010 | 0.963 | 0.001 | 0.001 |
| | RLL-CS | 1.025 | 1.014 | 0.994 | 0.018 | 0.017 |
| **conserve2nd** | CS-MPAS | 0.986 | 0.973 | 0.823 | -0.019 | -0.021 |
| | MPAS-RLL | 1.057 | 1.031 | 0.930 | 0.014 | 0.014 |
| | RLL-CS | 0.999 | 0.943 | 0.886 | -0.056 | -0.056 |

In order to better understand the relative accuracy of the first and second-order conservative remapping implementations in ESMF, we performed a comparative analysis for all grid combinations using standard global error norms. These results are shown in Figure 6, where the $\|E\|_{L_2}$ and $\|E\|_{H_1}$ error profiles for both the conservative schemes are compared as a function of remap iterations ($N_r$) for the CS-MPAS mesh combination. The computed error metrics from both the schemes indicate that, even though the convergence rates are similar, the 'conserve2nd' option in ESMF clearly produces a significantly better approximation for the remapped field, irrespective of the mesh resolution or field characteristics in the tested samples. It is also evident that the ESMF algorithms are more accurate for the RLL-CS mesh types in all global error norms, indicating sensitivity to the underlying element types in the mesh.



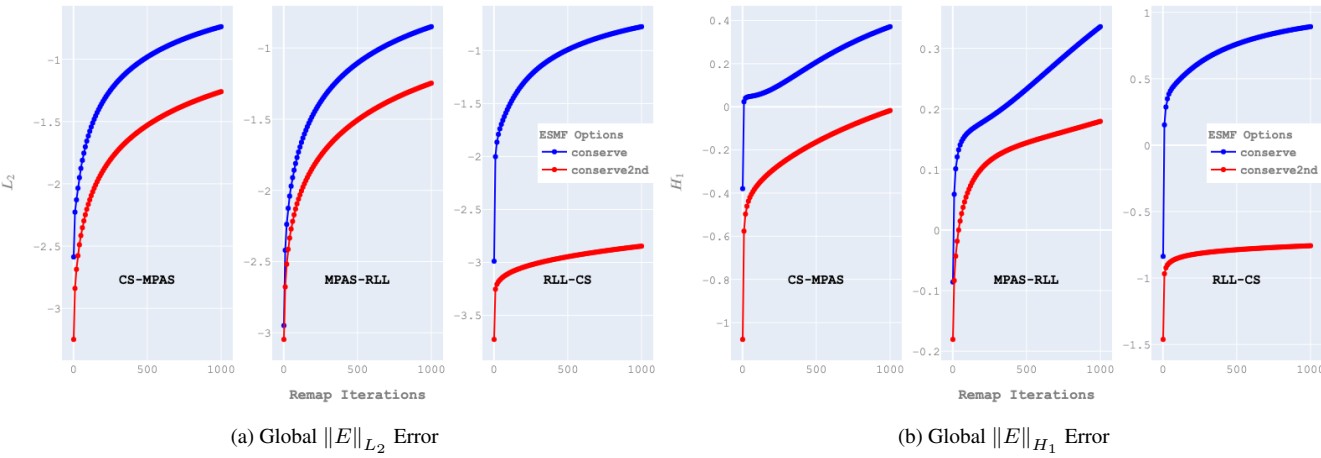

(a) Global $\|E\|_{L_2}$ Error           (b) Global $\|E\|_{H_1}$ Error

**Figure 6.** ESMF comparison of 'conserve' and 'conserve2nd' options for all mesh types and the TOPO field, using $\|E\|_{L_2}$ (left) and $\|E\|_{H_1}$ (right) global error metrics as function of $N_r$.

### 5.1.2 TempestRemap

The conservative high-order linear maps computed by TempestRemap, as shown in Table 2, produce higher-order expected theoretical convergence rates in comparison to ESMF for the smooth analytical solution fields. The convergence results presented here have been generated with a bi-degree-$p$ basis reconstruction using a rectangular truncation strategy that computes

5   source patches based on edge-adjacency graphs.

Note that even for smooth solutions, the convergence rates observed for $p > 2$ degrade during the conservative reconstruction for FV-FV maps. This degradation is evident in the rate reduction in all the global error norms for $p > 2$, which indicates that a larger patch size may be necessary to accurately and consistently recover the smooth field after remap from $\Omega_s$ to $\Omega_t$. The rate reduction is especially noticeable in the presence of pole singularities like that of RLL meshes, where the $\|E\|_{L_\infty}$ shows further

10   reduction in convergence rates. The failure to achieve $\mathcal{O}(h^4)$ or higher-order accuracy is an artifact of the implementation in TempestRemap, which can be improved with further analysis and is not a limitation of the underlying numerical method.





**Table 2.** TempestRemap: Convergence rates for ANALYTICALFUN1 field on all mesh types

| Degree | Grid Type | $\|E\|_{L_1}$ | $\|E\|_{L_2}$ | $\|E\|_{L_\infty}$ | $\|E\|_{H_1}$ | $|E|_{H_1}$ |
|---|---|---|---|---|---|---|
| | CS-MPAS | 1.031 | 1.022 | 0.857 | 0.004 | 0.002 |
| $p = 0$ | MPAS-RLL | 1.034 | 1.010 | 0.963 | 0.001 | 0.001 |
| | RLL-CS | 1.025 | 1.014 | 0.994 | 0.018 | 0.017 |
| | CS-MPAS | 2.096 | 2.056 | 1.882 | 1.069 | 1.067 |
| $p = 1$ | MPAS-RLL | 2.033 | 2.008 | 1.923 | 1.027 | 1.027 |
| | RLL-CS | 2.018 | 2.005 | 1.964 | 0.995 | 0.994 |
| | CS-MPAS | 3.096 | 3.101 | 2.900 | 2.105 | 2.103 |
| $p = 2$ | MPAS-RLL | 3.126 | 3.103 | 2.996 | 2.069 | 2.069 |
| | RLL-CS | 3.030 | 3.018 | 2.995 | 2.022 | 2.021 |
| | CS-MPAS | 3.258 | 3.298 | 3.125 | 2.347 | 2.346 |
| $p = 3$ | MPAS-RLL | 3.144 | 3.112 | 2.570 | 2.311 | 2.310 |
| | RLL-CS | 3.936 | 3.000 | 2.016 | 2.000 | 1.999 |

TempestRemap produces conservative solution projections between mesh combinations that can be 3rd-order with $p = 2$, for smooth fields. However, even if local element-wise conservation can be guaranteed in overlay-based high-order $L^2$ minimization schemes, monotonic reconstructions may not be strictly possible without additional effort. This behavior is because Godunov's theorem (Godunov, 1959) precludes the existence of optimal high-order linear maps that are also monotone. Due to this restriction, and since Gibbs phenomena (Jerri, 2013) are ubiquitous with high-order maps when steep gradients are encountered, global bounds can be preserved in TempestRemap by employing nonlinear filtering algorithms such as CAAS during the online solution transfer in ESMs.

### 5.1.3 GMLS

The remapping scheme based on the meshless Generalized Moving Least-Squares (GMLS) method demonstrates the flexibility to deliver higher-order convergence for scalar field data. The convergence rates computed for the ANALYTICALFUN1 field are shown in Table 3 for various polynomial degrees.



**Table 3.** GMLS and GMLS-CAAS: Convergence rates for ANALYTICALFUN1 field on all mesh types. Note that the superconvergence of gradients for $p = 4$ appears to be special for ANALYTICALFUN1; for ANALYTICALFUN2, GMLS converged at about fourth order, as theoretically expected, in $\|E\|_{H_1}$ and $|E|_{H_1}$.

| Degree | Grid Type | GMLS $\|E\|_{L_1}$ | $\|E\|_{L_2}$ | $\|E\|_{L_\infty}$ | $\|E\|_{H_1}$ | $|E|_{H_1}$ | GMLS-CAAS $\|E\|_{L_1}$ | $\|E\|_{L_2}$ | $\|E\|_{L_\infty}$ | $\|E\|_{H_1}$ | $|E|_{H_1}$ |
|---|---|---|---|---|---|---|---|---|---|---|---|
| $p=1$ | CS-MPAS | 1.957 | 1.957 | 1.944 | 1.838 | 1.832 | 1.957 | 1.957 | 1.944 | 1.838 | 1.832 |
| | MPAS-RLL | 1.990 | 1.990 | 1.978 | 1.663 | 1.652 | 1.990 | 1.990 | 1.978 | 1.663 | 1.652 |
| | RLL-CS | 1.981 | 1.980 | 1.979 | 1.870 | 1.864 | 1.981 | 1.980 | 1.979 | 1.870 | 1.864 |
| $p=2$ | CS-MPAS | 3.678 | 3.685 | 3.414 | 2.378 | 2.371 | 3.670 | 2.983 | 2.018 | 1.925 | 1.923 |
| | MPAS-RLL | 3.647 | 3.581 | 3.136 | 2.106 | 2.104 | 3.649 | 3.103 | 2.078 | 1.975 | 1.975 |
| | RLL-CS | 3.666 | 3.615 | 3.363 | 2.241 | 2.235 | 3.659 | 2.877 | 1.985 | 1.862 | 1.861 |
| $p=3$ | CS-MPAS | 3.982 | 3.982 | 3.978 | 3.954 | 3.952 | 3.924 | 3.118 | 2.022 | 1.948 | 1.944 |
| | MPAS-RLL | 3.986 | 3.986 | 3.978 | 3.815 | 3.808 | 3.969 | 3.321 | 2.180 | 2.001 | 2.000 |
| | RLL-CS | 3.985 | 3.985 | 3.985 | 3.969 | 3.968 | 3.914 | 2.975 | 2.035 | 1.873 | 1.872 |
| $p=4$ | CS-MPAS | 6.000 | 6.001 | 5.996 | 5.939 | 5.936 | 3.724 | 2.890 | 1.997 | 1.902 | 1.901 |
| | MPAS-RLL | 5.993 | 5.993 | 5.980 | 5.736 | 5.727 | 3.866 | 2.965 | 2.121 | 1.973 | 1.973 |
| | RLL-CS | 5.969 | 5.969 | 5.965 | 5.930 | 5.928 | 3.747 | 2.873 | 2.010 | 1.881 | 1.880 |

The convergence rates for the nominal GMLS scheme and the GMLS-CAAS remapping method with a post-processing step in Table 3 show that, in general, high-order accuracy can be achieved. However, using the nominal GMLS scheme for climate modeling problems can result in non-conservative and potentially oscillatory reconstructions for fields with strong gradients. Hence, GMLS-CAAS algorithm is especially advantageous to enable global and local bounds preservation. Note

that the augmented GMLS-CAAS algorithm suffers from convergence degradation for higher polynomial degree values and is limited to $\mathcal{O}(h^3)$ for this smooth field data in $\|E\|_{L_2}$. There is limited theoretical proof for convergence rates of CAAS filter in the literature (Bradley et al., 2019) for arbitrary problems. However, in two dimensions, one can consider a field with 1D connected band of extrema that demonstrates a maximum rate of convergence of $\mathcal{O}(h^3)$ in $\|E\|_{L_2}$ norm. Algorithms like CAAS function by clipping newly formed local extrema resulting from higher-order reconstructions and computing a redistribution of

the mass deficit accordingly. We hypothesize that the observed convergence degradation is primarily a result of these clipping and redistribution steps.

Even though the GMLS-CAAS remaps show lower convergence rates, it provides the benefit of making the scheme globally conservative and monotone. The CAAS algorithm requires runtime modification of the projected fields to ensure global and local bounds, and the nonlinear solution-dependent filter can eliminate Gibbs oscillations, providing better stability during

remap operation.





### 5.1.4 WLS-ENOR

The convergence analysis for the WLS-ENOR scheme shows that even for high polynomial degrees, theoretically expected rates are observed for the smooth analytical field. The convergence rates for various degrees are tabulated in Table 4.

**Table 4.** WLS-ENOR: Convergence rates for ANALYTICALFUN1 field on all mesh types

| Degree | Grid Type | $\|E\|_{L_1}$ | $\|E\|_{L_2}$ | $\|E\|_{L_\infty}$ | $\|E\|_{H_1}$ | $|E|_{H_1}$ |
|--------|-----------|---------------|---------------|--------------------|----------------|--------------|
| | CS-MPAS | 3.039 | 3.044 | 2.854 | 2.063 | 2.061 |
| $p=2$ | MPAS-RLL | 3.006 | 3.003 | 2.903 | 2.002 | 2.002 |
| | RLL-CS | 3.030 | 3.036 | 2.931 | 2.046 | 2.045 |
| | CS-MPAS | 4.989 | 4.991 | 4.963 | 3.993 | 3.991 |
| $p=3$ | MPAS-RLL | 5.023 | 5.021 | 4.932 | 3.997 | 3.997 |
| | RLL-CS | 4.961 | 4.981 | 4.887 | 3.989 | 3.988 |
| | CS-MPAS | 5.131 | 5.145 | 4.845 | 4.141 | 4.139 |
| $p=4$ | MPAS-RLL | 5.007 | 5.001 | 4.853 | 3.997 | 3.997 |
| | RLL-CS | 5.170 | 5.118 | 4.951 | 4.033 | 4.032 |

We note that the WLS-ENOR algorithm is equipped with an internal nonlinear filtering (or more precisely, mollification) mechanism to detect sharp gradients and discontinuities, in order to adaptively choose the weights during the high-order reconstruction process locally. In contrast to the GMLS-CAAS high-order meshless scheme with a post-processing filter that results in a convergence order degradation, the WLS-ENOR scheme remains consistently accurate for smooth functions up to $p = 4$ in our experiments.

### 5.1.5 Real Fields: Convergence Rate Comparisons

While high-order convergence rates are achievable for smooth field profiles of ANALYTICALFUN1 and ANALYTICALFUN2, maintaining theoretical rates of convergence for 'real-world' field data is not guaranteed, because such data may lack the regularity necessary to achieve the best possible rates. Due to the presence of strong $\mathcal{C}^0$ and $\mathcal{C}^1$ discontinuities in the real scalar fields, which are representative of flux fields exchanged between components in coupled climate solvers, a remap comparison on the SPH sampled TPW field data was computed, as tabulated in Table 5. The convergence rate comparisons using different remap schemes show that the computed rates for higher-order polynomial reconstructions fall severely short of theoretical rates as expected, relative to the rates shown previously for ANALYTICALFUN1.

However, it can be observed that higher than first-order schemes tend to show better behavior on all meshes and fields tested. Additionally, the linear maps computed with ESMF underperform the other schemes, as evident from the gradient error norms, $\|E\|_{H_1}$ and $|E|_{H_1}$. The meshless schemes are competitive in terms of accuracy bounds and provide a viable alternative





approach for usage in production climate simulations, where traditionally $L^2$-minimization schemes and low-order bilinear maps have been routinely used.

**Table 5.** Comparing global error measures for the TPW field with different remap schemes on all mesh types of finest $\Omega_s$ and $\Omega_t$ resolutions

| Grid Type | Remapping Scheme | $\|E\|_{L_1}$ | $\|E\|_{L_2}$ | $\|E\|_{L_\infty}$ | $\|E\|_{H_1}$ | $|E|_{H_1}$ |
|---|---|---|---|---|---|---|
| **CS-MPAS** | ESMF ('conserve2nd') | 1.209 | 1.180 | 0.881 | 0.132 | 0.128 |
| | TempestRemap ($p = 3$) | 1.281 | 1.307 | 1.396 | 1.375 | 1.384 |
| | GMLS ($p = 4$) | 1.761 | 1.834 | 2.129 | 2.515 | 2.551 |
| | GMLS-CAAS ($p = 4$) | 1.785 | 1.896 | 2.063 | 2.258 | 2.264 |
| | WLS-ENOR ($p = 4$) | 1.677 | 1.810 | 2.261 | 2.517 | 2.553 |
| **MPAS-RLL** | ESMF ('conserve2nd') | 1.216 | 1.133 | 1.004 | 0.056 | 0.056 |
| | TempestRemap ($p = 3$) | 2.233 | 2.204 | 2.132 | 2.156 | 2.148 |
| | GMLS ($p = 4$) | 2.233 | 2.208 | 2.230 | 2.265 | 2.274 |
| | GMLS-CAAS ($p = 4$) | 2.238 | 2.218 | 2.073 | 1.952 | 1.946 |
| | WLS-ENOR ($p = 4$) | 2.242 | 2.223 | 2.708 | 2.767 | 2.797 |
| **RLL-CS** | ESMF ('conserve2nd') | 1.299 | 1.242 | 1.010 | -0.018 | -0.021 |
| | TempestRemap ($p = 3$) | 1.353 | 1.327 | 1.114 | 1.097 | 1.081 |
| | GMLS ($p = 4$) | 1.353 | 1.364 | 1.499 | 1.739 | 1.768 |
| | GMLS-CAAS ($p = 4$) | 1.377 | 1.432 | 1.817 | 1.818 | 1.827 |
| | WLS-ENOR ($p = 4$) | 1.357 | 1.343 | 1.727 | 1.728 | 1.758 |

Note that in this particular comparison, we selected the highest polynomial degree $p$ that was tested for each remapping method, even though for production climate simulations, typically $p = 1$, or $p = 2$ may yield a more numerically stable solution due to Gibbs phenomena. In such operational circumstances, high-order methods for TempestRemap and GMLS should be augmented with nonlinear post-processing filters, such as CAAS.

### 5.2 Global Conservation

All mesh-based $L^2$ projection scheme implementations chosen in this study, namely ESMF and TempestRemap, are globally conservative by the nature of the underlying numerics (Ullrich and Taylor, 2015). Since these implementations compute consistent reconstructions using overlay meshes, any deficit with respect to exact conservation can be verified by performing column sum operations on the discrete linear map matrix representing the projection or mass-matrix operator.

The global field integral error indicating conservation deficit for TPW field variable is shown in Figure 7 for different mesh resolutions of the CS-MPAS mesh type. In these plots, the y-axes represent the logarithmic error of the global integral value between solution remapped onto $\Omega_t$ relative to the reference solution sampled on $\Omega_t$. Hence, lower values indicate better satisfaction of the global conservation metric.



The expected behavior of preserving global integrals in the remapped solutions, as illustrated in Figure 7, have been verified for both low and higher-order maps, irrespective of the mesh topology, or the field variables being transferred across meshes. Notably, the slow accumulation of roundoff error in the global field integral as a function of remap iterations has a detrimental effect to global conservation. This effect is seen in the field projections computed with TempestRemap through two sparse

5    matrix-vector (SpMV) products in each remap iteration and is distinctly subdued in the low-order ESMF map projections, especially at fine mesh resolutions.

(a) Resolution: CS(0)-MPAS(4)

(b) Resolution: CS(4)-MPAS(0)

(c) Resolution: CS(4)-MPAS(4)

**Figure 7.** Global conservation metric for the TPW field on CS-MPAS meshes with different resolutions in semi-log scale

The WLS-ENOR scheme achieves excellent global conservation by applying adaptive quadrature rules to integrate the functions to (near) machine precision and redistributing the conservation errors locally near discontinuous regions. The unmodified GMLS scheme is non-conservative without any post-processing and hence is not presented here. However, using the nonlinear

10    CAAS filtering algorithm with the GMLS remapping scheme provides global conservation to user-specified tolerances.





This conservation metric describes whether the remapped solution preserves the global integral over the domain, which is often a desirable constraint for most scalar variables in order to ensure that the total mass and energy in the closed simulation system remain constant. However, for some scalar fields, such a conservation constraint may not be strictly mandated, and this enables the use of even the non-conservative GMLS scheme, which has been demonstrated to achieve excellent accuracy.

## 5.3  Monotonicity - Global Extrema Preservation

Departures away from global extrema, i.e., overshoots and undershoots, provide a useful metric to assess the stability of the remap operator under investigation. A perfectly stable method has a zero value for this metric. The results for the monotonicity metric as a function of the remap iterations are shown in Figure 8 and Figure 9 for $L_{max}$ of CFR variable on MPAS-RLL, and $L_{min}$ of the TOPO variable on CS-MPAS meshes, respectively. These field variables have been specifically chosen to showcase the effects of the remapping algorithms on the well-defined upper and lower global field bounds that cannot be violated.

The results clearly demonstrate that the mesh-based remapping schemes do not adhere to the global extrema (both maxima and minima) after the first linear remap operator application. The magnitude of this departure is resolution dependent, but the behavior is consistent in all cases tested. The use of low-order, dissipative linear maps from ESMF shows a slow drift away from global maxima, which are recovered after several remaps (around 700 in Figure 8). However, the global minima increase continuously within the bounds of the reference solution. Hence, the 'conserve2nd' option in ESMF damps the global maxima and amplifies the global minima.

In contrast, the high-order TempestRemap solutions show a drift away from the bounds in every case. The mildly damped increase of the metric as a function of remap operator application signals the presence of spurious high-frequency modes in the linear map, which is more prevalent as the degree of reconstruction increases. Obtaining high-order remapped solutions in addition to preserving monotonicity in these schemes will require post-processing filters such as CAAS, which can be used to improve the behavior in TempestRemap for such problems of interest.

Compared to the traditional remapping schemes, the meshless and hybrid remap schemes, namely GMLS-CAAS and WLS-ENOR, maintain the global bounds for all test cases and fields tested in this study. Augmenting the non-monotone GMLS scheme with CAAS bounds preservation algorithm shows excellent, stable behavior for all cases. WLS-ENOR shows relatively high compliance to actual bounds because its non-oscillatory weights near discontinuities essentially preserve the monotonicity and convexity of the solutions. However, when the meshes are sufficiently fine, the functions may appear smooth relative to the grid resolution, and in such scenarios, WLS-ENOR would not apply the adaptive weights and the global extremes may oscillate randomly at a magnitude comparable to the local discretization error, as seen in Figure 8(b).

## 5.4  Locality - Local Extrema Preservation

Similar to the global bounds metrics that measure solution monotonicity, the local bounds metrics compute the norm of error resulting from the comparison of the reference sampled field data against the projected field in a one-ring local neighborhood. This metric provides insight into the preservation of local field features on repeated remap operator applications. The com-

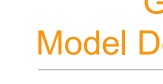
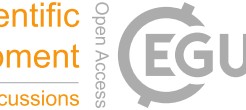


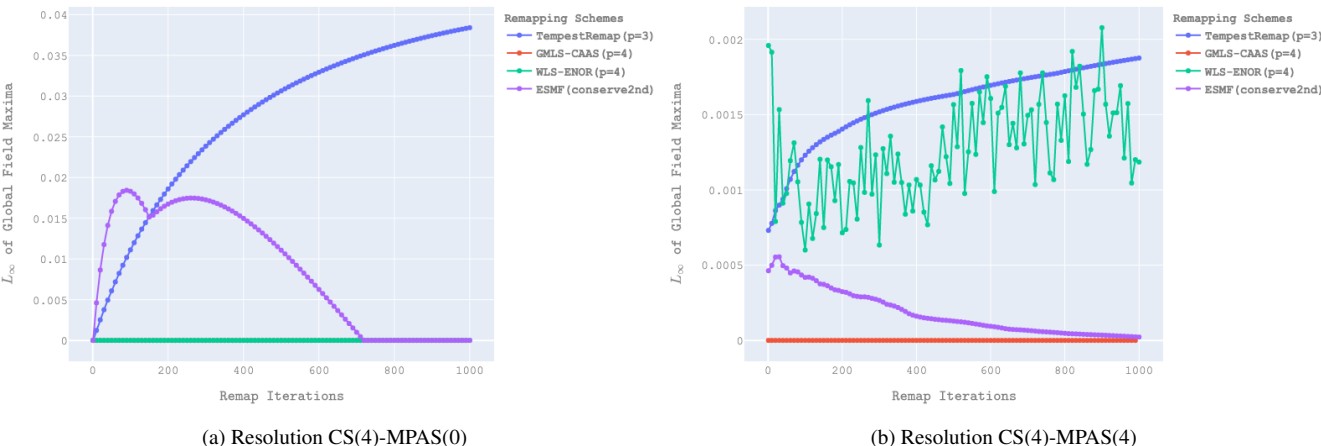

(a) Resolution CS(4)-MPAS(0)        (b) Resolution CS(4)-MPAS(4)

**Figure 8.** $L_{max}$ metric for the CFR field on MPAS-RLL with different target mesh resolutions

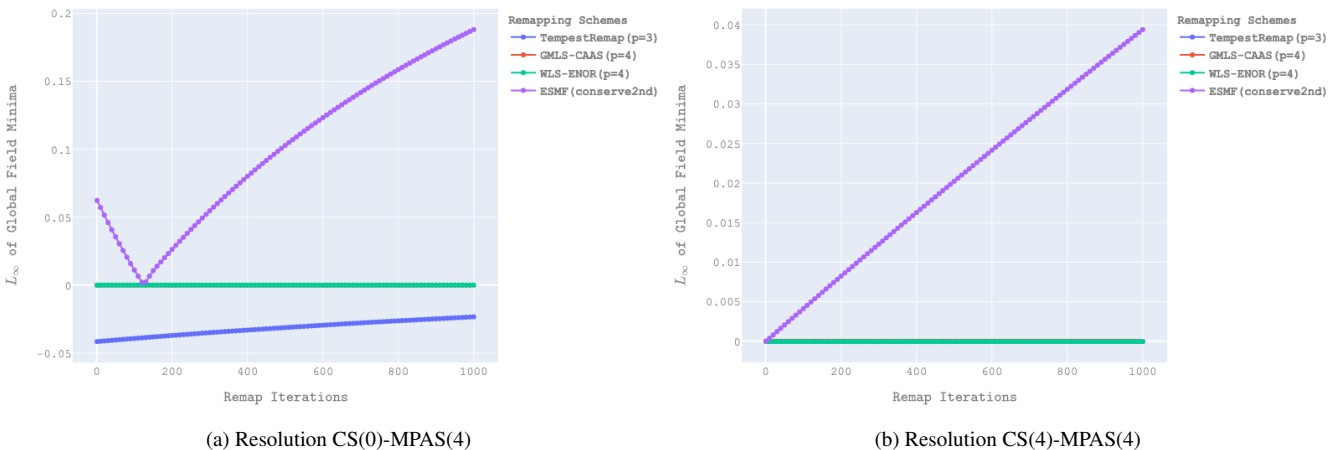

(a) Resolution CS(0)-MPAS(4)        (b) Resolution CS(4)-MPAS(4)

**Figure 9.** $L_{min}$ metric for the TOPO field on CS-MPAS with different source mesh resolutions

puted locality metrics for maxima and minima for various mesh resolutions and real-world fields are shown in Figure 10 and Figure 11.

These results showcase that the high-order TempestRemap implementations perform quite well with minimal feature loss for all fields on all meshes. In contrast, the low-order ESMF linear maps have high dissipation growth that is dependent on

5    both the resolution and remap iterations. For high-resolution meshes, the amplification of the dissipation at every iteration can even be undamped and grows linearly with the ESMF 'conserve2nd' option. This result implies that as the number of remap iterations increase, more information on local features is lost during every transfer for these cases.





For the WLS-ENOR scheme, which uses a hybrid strategy to detect local features and adaptive quadrature rules to resolve a high discrepancy in mesh resolutions, its preservation of local bounds exhibits a strong dependence on the spatial resolution of $\Omega_s$ and $\Omega_t$. However, unlike ESMF, the slope of metric indicating departure away from local bounds remains near zero as remap iterations are increased, signifying that the reconstructions remain stable on repeated applications without any further

5    feature losses, probably due to its use of adaptive quadrature rules.

In the meshless remapping category, the GMLS schemes utilize the CAAS algorithm to clip and conserve fields in a local subset of the neighborhood for each evaluation point. For GMLS-CAAS, local bounds must be given to CAAS, and it is important to consider how they are determined. As GMLS is a meshless solution technique, local bounds for each site of reconstruction are determined by computing the maximum and minimum of the values in the neighborhood (determined meshlessly

10    using a K-d tree) used for reconstruction from $\Omega_s$. With a coarse $\Omega_s$ resolution, these local bounds computed from K-d tree queries are no longer accurate estimators to enforce tight bounds in a 1-ring neighborhood of $\Omega_t$, which is the metric measured in this study for local bounds preservation. The discrepancy in bounds for such scenarios could be minimized by using a mesh intersection of the cells in the source and target neighborhoods, but no attempt was made to do this for GMLS-CAAS in keeping with the spirit of its application as a purely meshless technique.





(a) Resolution CS(0)-MPAS(4)

(b) Resolution CS(4)-MPAS(0)

(c) Resolution CS(4)-MPAS(4)

**Figure 10.** $L_{max,2}$ metric for the TPW field on CS-MPAS meshes with different resolutions

Lauritzen and Nair (2008) noted that when remapping using their monotone and conservative CaRS algorithm, the higher-order reconstructions do not significantly improve accuracy when $\Omega_s$ is finer than $\Omega_t$. However, the reverse case shows significant benefit with high-order polynomial reconstructions. These conclusions are similar to those observed in the current intercomparison experiments shown in Figure 10 and Figure 11.





(a) Resolution RLL(0)-CS(4)

(b) Resolution RLL(4)-CS(0)

(c) Resolution RLL(4)-CS(4)

**Figure 11.** $L_{min,2}$ metric for the TOPO field on RLL-CS meshes with different resolutions

One of the key outcomes of this analysis indicates that using nonlinear filtering algorithms like CAAS can provide the benefits of property preservation to achieve global conservation and monotonicity constraints. However, these advantages can be offset by the higher local dissipation effects, especially on disparate mesh resolutions of $\Omega_s$ and $\Omega_t$. So depending on the problem use-case for which linear maps are being used to transfer solution fields between meshes with varying resolutions, an adaptive approach may be taken to choose a nonlinear filter when monotonicity constraints are important and to apply it with care, especially for high-order maps where feature preservation may suffer.

## 5.5 Analysis on RRM Meshes

We restrict the analysis in this study to all the conservative variations of the schemes chosen. As a result, the global conservation metric is truly satisfied for all cases tested with RRM meshes. The following subsections provide a detailed comparison of the error resolutions and global/local bound preservation metrics.

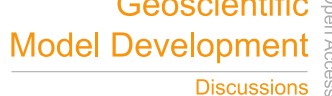



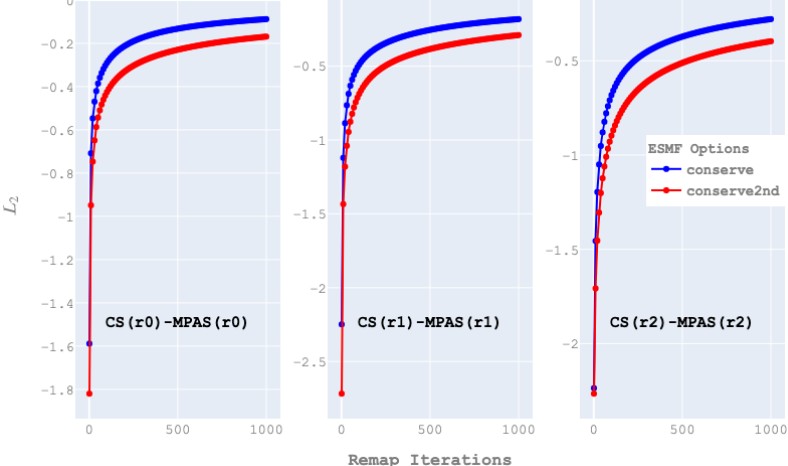

**Figure 12.** ESMF comparison of 'conserve' and 'conserve2nd' options for the TPW field on all CS-MPAS RRM meshes

### 5.5.1 Error Metrics Comparison

In order to compare the performance of the different remapping schemes on more realistic and complex unstructured RRM meshes, the global error norms, $\|E\|_{L_1}$, $\|E\|_{L_2}$, $\|E\|_{L_\infty}$, $\|E\|_{H_1}$ and $|E|_{H_1}$ are computed and tabulated in Table 6. These results were computed on the finest CS-MPAS RRM mesh combination to look at the accurate resolution of different fields. The global errors with respect to all error norms are considerably smaller in WLS-ENOR and TempestRemap for all the field variables sampled on the RRM input meshes.

In contrast, the low-order ESMF implementation results in a larger error comparable to GMLS-CAAS with $p = 4$ for all the real fields, even though the relative accuracy on smooth analytical fields is much better with GMLS-CAAS. The use of ESMF with 'conserve2nd' option and the GMLS-CAAS schemes deliver larger $\|E\|_{H_1}$ and $|E|_{H_1}$ errors than the more accurate WLS-ENOR and TempestRemap remapped solutions. This summary shows that the state-of-art ESMF conservative remapping schemes can be complemented by other algorithms that have relatively better accuracy profiles for many coupled climate modeling problems.

In order to make a fair selection of ESMF conservative algorithm, the comparison between the 'conserve' and 'conserve2nd' implementations was repeated on RRM meshes. The $\|E\|_{L_2}$ error profiles for various equal resolution source and target meshes are shown in Figure 12 for the TPW variable. The results suggest that the behavior of 'conserve2nd' option remains marginally better than the 'conserve' option in these experiments, even though the magnitude of accuracy gain with the 'conserve2nd' is not as attractive in comparison to the case of regularly refined meshes analyzed in Figure 6. An outcome of this experiment is that the results clearly demonstrate the superiority of the 'conserve2nd' conservative remapping implementation option in ESMF in comparison to the 'conserve' option, and is hence recommended for use in all existing production climate simulations when high-order maps are unavailable.



**Table 6.** Comparison of global error norm metrics for different remap schemes on the finest CS-MPAS RRM mesh combination, ordered from best to worst in terms of $\|E\|_{L_2}$ error for all field variables

| Variable | Method | $\|E\|_{L_1}$ | $\|E\|_{L_2}$ | $\|E\|_{L_\infty}$ | $\|E\|_{H_1}$ | $\|E\|_{H_1}$ |
|---|---|---|---|---|---|---|
| **ANALYTICALFUN1** | WLS-ENOR ($p=4$) | 5.457e-12 | 6.057e-12 | 1.418e-11 | 1.563e-06 | 1.563e-06 |
| | TempestRemap ($p=3$) | 3.410e-09 | 6.138e-07 | 9.785e-05 | 4.387e-01 | 4.387e-01 |
| | GMLS-CAAS ($p=4$) | 1.129e-04 | 1.180e-04 | 1.709e-04 | 4.945e+00 | 4.945e+00 |
| | ESMF ('conserve2nd') | 1.475e-04 | 2.031e-04 | 1.288e-03 | 9.610e+01 | 9.610e+01 |
| **ANALYTICALFUN2** | WLS-ENOR ($p=4$) | 5.737e-13 | 7.329e-13 | 4.114e-12 | 2.066e-07 | 2.066e-07 |
| | TempestRemap ($p=3$) | 4.692e-10 | 9.353e-08 | 2.268e-05 | 6.693e-02 | 6.693e-02 |
| | GMLS-CAAS ($p=4$) | 1.226e-05 | 1.502e-05 | 3.306e-05 | 6.641e-01 | 6.641e-01 |
| | ESMF ('conserve2nd') | 2.223e-05 | 3.344e-05 | 3.144e-04 | 1.543e+01 | 1.543e+01 |
| **Total Precipitable Water (TPW)** | WLS-ENOR ($p=4$) | 1.008e-03 | 9.474e-04 | 7.281e-04 | 1.405e+01 | 1.405e+01 |
| | TempestRemap ($p=3$) | 1.008e-03 | 9.474e-04 | 7.281e-04 | 1.406e+01 | 1.406e+01 |
| | ESMF ('conserve2nd') | 1.009e-03 | 9.527e-04 | 1.127e-03 | 4.300e+01 | 4.300e+01 |
| | GMLS-CAAS ($p=4$) | 1.016e-03 | 1.029e-03 | 1.264e-03 | 5.879e+01 | 5.879e+01 |
| **Cloud Fraction (CFR)** | TempestRemap ($p=3$) | 2.802e-04 | 3.648e-04 | 5.248e-03 | 7.254e+01 | 7.254e+01 |
| | ESMF ('conserve2nd') | 3.042e-04 | 4.110e-04 | 5.490e-03 | 1.030e+02 | 1.030e+02 |
| | WLS-ENOR ($p=4$) | 2.993e-04 | 4.689e-04 | 7.082e-03 | 1.300e+02 | 1.300e+02 |
| | GMLS-CAAS ($p=4$) | 6.496e-04 | 8.577e-04 | 6.930e-03 | 1.627e+02 | 1.627e+02 |
| **Global Topography (TOPO)** | WLS-ENOR ($p=4$) | 5.672e-03 | 5.407e-03 | 3.714e-03 | 1.830e+02 | 1.830e+02 |
| | TempestRemap ($p=3$) | 5.672e-03 | 5.407e-03 | 3.714e-03 | 1.830e+02 | 1.830e+02 |
| | ESMF ('conserve2nd') | 5.683e-03 | 5.424e-03 | 3.933e-03 | 2.572e+02 | 2.572e+02 |
| | GMLS-CAAS ($p=4$) | 6.149e-03 | 6.023e-03 | 5.262e-03 | 4.035e+02 | 4.035e+02 |

### 5.5.2 Monotonicity Metrics

Similar to the uniformly refined cases, the monotonicity metric $L_{max}$ was analyzed on the finest resolutions of the CS-MPAS regionally refined meshes for the CFR field, as plotted in Figure 13. All of the remapping schemes tested in this experiment maintain strict bounds on global maxima and minima at the end of the remapping iterations, except for the high-order Tempest-
5  pestRemap algorithm. The observation shown here confirms that TempestRemap requires additional post-processing monotone filters such as CAAS during the online solution projection step in a simulation, in order to ensure strict global bounds preservation as a function of the remap iterations.

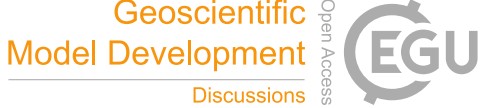

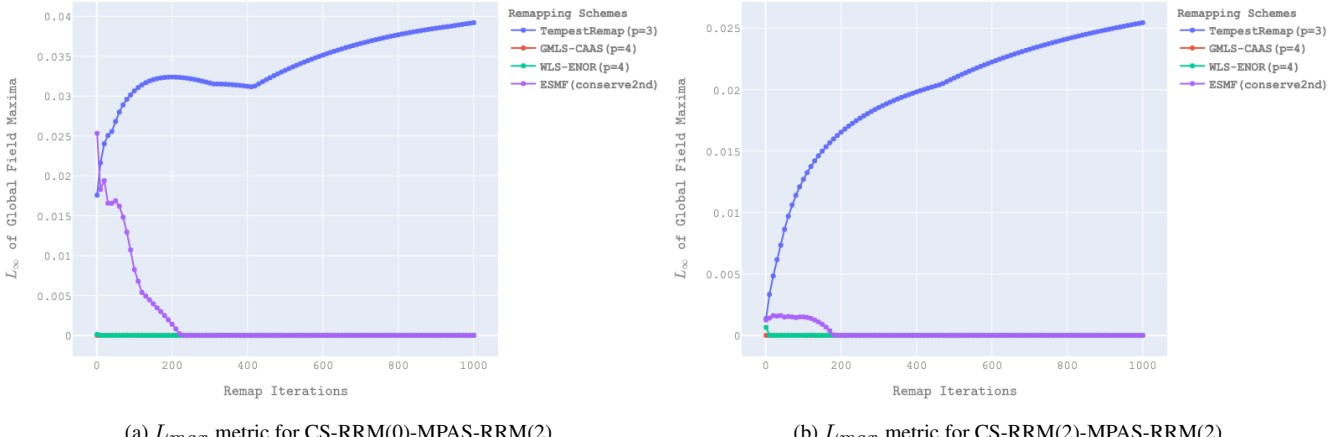

(a) $L_{max}$ metric for CS-RRM(0)-MPAS-RRM(2)      (b) $L_{max}$ metric for CS-RRM(2)-MPAS-RRM(2)

**Figure 13.** $L_{max}$ metric identifying departure from global maxima for CFR field for coarse to fine, and similar refinements of CS and MPAS RRM meshes

### 5.5.3 Locality Metrics

The observed trends in the locality metrics computed on RRM meshes are similar to those exhibited in the uniformly refined mesh experiments presented earlier. Hence, for brevity, only the $L_{max,2}$, and $L_{min,2}$ metrics are presented for the TOPO field on the finest resolution of RRM meshes in Figure 14.

It is imperative to recognize that the high order remapping schemes from TempestRemap and WLS-ENOR continue to generate minimally diffusive projections on target RRM meshes. This behavior is quite attractive as local features in the fields, even in the presence of strong gradients, are resolved accurately with very low dissipation away from the sampled reference field data. The locally adaptive, discontinuity tracking WLS-ENOR algorithm shows strongly stable behavior for all test cases.

However, the meshless GMLS-CAAS scheme and the low-order ESMF scheme exhibit severe departure away from these

10 local bounds that are consistent with the relatively larger $\|E\|_{H_1}$ and $|E|_{H_1}$ observed in Table 6. Depending on the solution fields being remapped between model components, such strong dissipation in sharp features could lead to local numerical artifacts in the coupled climate system.

One key observation from the global monotonicity metrics shown in Figure 13 is that TempestRemap fails to maintain strict global bounds, while low-order ESMF and the meshless GMLS-CAAS schemes recover them consistently. However, the local

bound-preservation metrics in Figure 14 demonstrate that when using the conservative, high-order TempestRemap algorithm, element-wise dissipation in the field data is strictly bounded and relatively much smaller in comparison to other remapping algorithms.



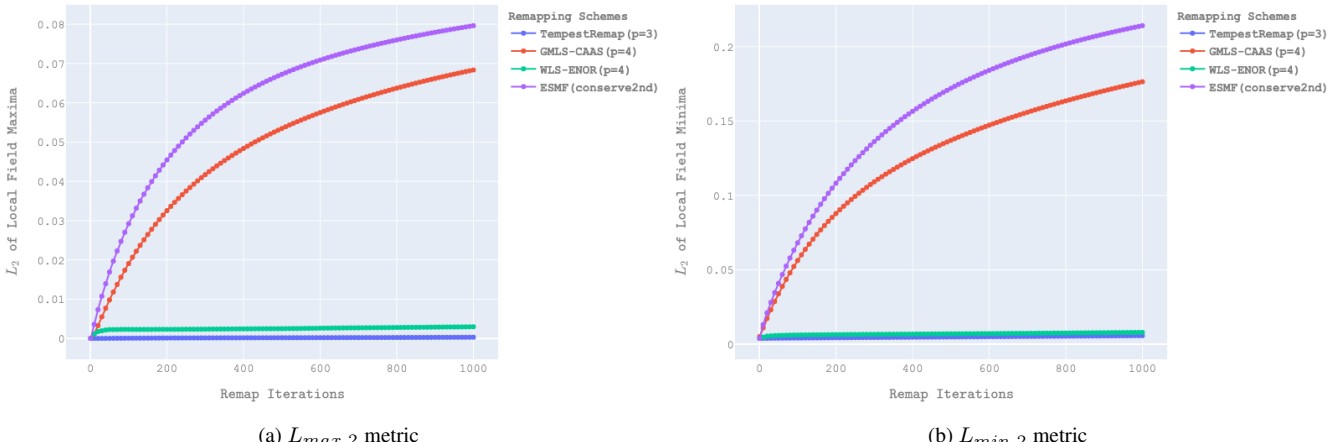

(a) $L_{max,2}$ metric  (b) $L_{min,2}$ metric

**Figure 14.** $L_{max,2}$ and $L_{min,2}$ metric for TOPO field on the finest refinements of CS and MPAS RRM meshes

## 6  Future Research Directions

This work provides a foundation for the systematic computation of key metrics of interest in regridding problems and implements the infrastructure to sample analytical and real-world fields and measure the metrics through a flexible Python code. In the current manuscript, we have presented detailed experiments to compare remapping schemes for FV-FV discretization settings on three specific mesh families (CS, MPAS, RLL) of varying resolutions, both uniform and regionally refined. There are several research directions that will be explored further in order to strengthen and improve the intercomparison protocol and apply it to more realistic climate science problems.

Four key extensions of interest in this context are:

1. Complex meshes, including successive $r-$adapted meshes, and realistic samples with topological holes

2. Complex discretizations, other than just FV for source and target components

3. Complex field descriptions, including vector data and preservation of nonlinear correlations in multiple fields

4. Computational efficacy, comparing numerical accuracy and algorithmic performance on next-generation architectures

Further details regarding these research directions are provided in the following subsections.

### 6.1  Complex Meshes: More than just global meshes

Uniformly refined meshes provide a simple infrastructure to test the consistency of remapping schemes, and additionally, regionally refined global grids help identify any spatial-scale dependencies in the algorithms. However, it is important to note that the meshes tested do not include any topological holes that are typical in production climate simulations. A direct





extension of the current study would be to perform FV-FV remapping studies on realistic meshes, especially with the MPAS ocean polygonal meshes (Petersen, 2018) that do not cover the entire sphere.

Several potential issues may arise when remapping fields on these complex topological meshes. For example, dealing with narrow isthmus regions like Panama, which is also at the boundary between ocean basins, or along the coast of Peru where

sea-level with high-altitude points from the Andes can influence and contaminate remaps to produce biases that appear along these coastal regions. Such regions can yield incorrect field remaps when using simple nearest-neighbors maps, high-order reconstructions, or even when computing the overlay mesh, as the numerical tolerances for querying neighborhood data become an important influencing factor in the overall accuracy.

The metrics definitions for standard error norms, global maxima, and global minima introduced in Section 3 are valid for

general meshes. But, in order for the global conservation metric $L_g$ shown in Equation (11) to be valid, the target domain has to fully envelop the source domain. Additionally, the definition and implementation of local dissipation metrics use a small neighborhood (one-ring patch) around the target cell, which would also impose a similar requirement. This implies that for analysis with realistic meshes, the metrics for evaluating the quality of solution field data transferred between the atmosphere and ocean components will strictly be measured only on the global atmosphere meshes to satisfy the above constraints. Note

that the input specification for such cases would require at least the area fractions and appropriate masks to be provided in order to enable efficient remap computations using the meshless and hybrid schemes like GMLS and WLS-ENOR.

Furthermore, another test study of interest would be to develop a sequence of meshes that slowly deform from the original mesh into some intermediary that resolves moving solution fields like tracer transport, eventually returning back to the original mesh. Such a test is referred to as 'cyclic-remap' test suites in the ALE community (Bochev and Shashkov, 2005). Performing

successive remap over the sequence of such meshes, one can directly compare the final resulting field after multiple remaps directly with the initial field without a need for a reference analytical solution field. Devising such a study requires mesh-based adaptivity and smoothing algorithms to be used effectively in addition to mesh optimization strategies to avoid element inversions (Brewer et al., 2003). On the other hand, any errors introduced due to SPH sampling functionals and discontinuities introduced due to clipping field bounds can be completely avoided with such a setup.

**6.2   Complex Discretizations: More than just Finite Volume**

The current study focused primarily on element averaged field data that is typical of FV discretization of models. Although many atmosphere and ocean models define the scalar fields as element-averaged piecewise constant data, it is imperative to extend the above analysis framework to Spectral-Element (SE) data defined on CS meshes so that it can be remapped onto either FV data on MPAS or RLL meshes or SE data on CS meshes with different resolutions. These extensions would allow

us to verify the flexibility of the high-order, conservative remapping schemes tested in the current study under discretization specifications commonly used in climate components like HOMME (Taylor et al., 2007). We also make a note that such extensions to more complex discretizations of source and target data should not require any modifications in the metrics definitions already presented in this manuscript. Further extensions to meshes that contain topological holes will still require $\Omega_s \subset \Omega_t$.





### 6.3 Complex Fields: More than just scalars

During the remapping of climate data in production runs and scientific analysis, it is important to preserve not only scalar fields but also vector fields and derived properties that provide better insights from simulations. For example, to understand the atmospheric flows, analysis of global wind patterns is often performed in weather prediction, which requires the evaluation of

derivatives or integrals of the vector wind velocity fields (Nair and Jablonowski, 2008). Some critical considerations in such scenarios require preservation of divergence-free velocity fields, conservation of vorticity, and preservation of wind-stress curl fields during the remap phase. Another example is, during the computation of tracer advection, certain tracers that contribute to an equation-of-state must be remapped consistently with the density field in order to preserve derived quantities. Typically, such vector fields are treated as collections of unrelated scalar fields that are remapped independently. However, such approaches

are deficient in preserving divergence-free conditions and can be inconsistent, since the propagation of remapping error in the components are not correlated.

Remapping algorithms based on mimetic schemes (Pletzer and Hayek, 2019) that provide exact conservation for both scalar and vector fields are promising in this direction. To our knowledge, existing remapping algorithms based on $L^2$ minimization and high-order interpolation-based algorithms need further research for tackling vector field data in problems of interest.

Furthermore, field data with nonlinear correlations such as those analyzed by (Lauritzen and Thuburn, 2012), adapted for remapping scenarios, can also be valuable in determining whether the solution transfer algorithms can remain conservative and preserve correlation properties. Nonlinear remapping schemes (Carey et al., 2001; Bochev and Shashkov, 2005; Bochev et al., 2011) may also prove to be viable options for these cases, albeit computational cost can be relatively much higher than using linear maps on decoupled scalar components. Additionally, some remapping metrics definitions introduced in Section 3 will

have to be extended for vector field data.

### 6.4 Computational Efficacy: Comparing accuracy and efficiency

The additional characteristic dimension that is essential to recognize in intercomparison studies is that the overall cost to obtain the remaps over the simulation cycle includes both a one-time setup cost, and a constantly growing computational load at every coupling step when field data need to be transferred between components. Hence, in order to better describe the computational

cost of the remapping algorithms, we could split the effort into an 'offline' cost and an 'online' cost. Note that often, what is sacrificed in terms of computational performance, is gained in the quality of the remapped solution through higher accuracy, global bounds preservation, and strong local feature resolution with minimal dissipation. Since the numerical efficiency and time to compute the solution are competing factors, the usage of advanced remapping algorithms for realistic cases will require a more detailed analysis of the computational complexity at scale.

The traditional mesh-based conservative remapping algorithms used in ESMF and TempestRemap require computation of an intersection or supermesh, $\Omega_{s \cap t}$, which in general is computationally expensive. While linear-complexity strategies like advancing front methods do exist in libraries like PANG (Gander and Japhet, 2013), and MOAB (Mahadevan et al., 2020) to compute mesh intersections, they can suffer from robustness issues when $\Omega_s$ has topological holes. An alternative approach is to





utilize a Kd-tree datastructure to accelerate queries on the meshes, in order to exactly locate the element containing a particular point, which is a fundamental operation in intersection mesh computation (Jansen et al., 1992). The leading computational cost for this operation is $\mathcal{O}(Nlog(N))$, where $N$ is the total number of elements in the mesh. The linear map computation itself follows typical FV or FE operator assembly workflows, and does remain bounded. The online cost of such a linear map

application is essentially then provided by the sparse matrix-vector (SpMV) operation. The complexity for SpMV is, however, dependent on the degree $p$ used for field reconstruction as it dictates the bandwidth of coupling between degrees-of-freedom in $\Omega_s$ and $\Omega_t$. The illustration in Figure 15 for ESMF ('conserve'), and TR ($p = 3$) remapping algorithms shows the number of CS elements required to reconstruct the solution on a single target MPAS element when computing projections between the CS(3)-MPAS(3) uniform refinement case. While the low-order ESMF algorithm may require only *2 FLOPS* per reconstruction

on a target element, TempestRemap ($p = 3$) would require a minimum of *50 FLOPS* to achieve better accuracy. To overcome some of these computational issues for large meshes, the use of MPI parallelism has been exploited in ESMF, and has been proven scalable with MOAB-TempestRemap (Mahadevan et al., 2020) libraries.

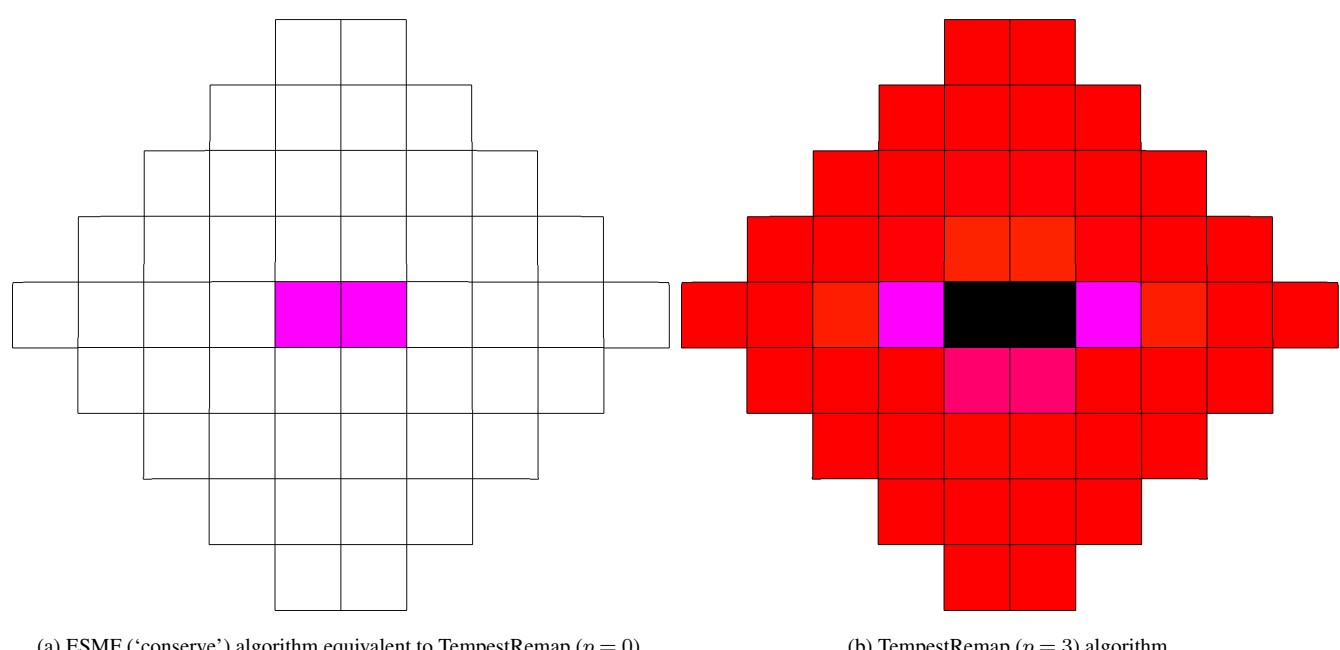

(a) ESMF ('conserve') algorithm equivalent to TempestRemap ($p = 0$)  (b) TempestRemap ($p = 3$) algorithm

**Figure 15.** Comparison of source field reconstructions showing DoF coupling in linear maps when utilizing different remapping algorithms and degrees of expansion.

Computation for the GMLS-CAAS meshfree approach on a manifold is dominated by the internal QR with pivoting factorization, requiring $\mathcal{O}\left(\frac{(p^2)^3}{6} \cdot N\right)$ *FLOPS* for the offline stencil calculation, where $p$ is the polynomial degree of the basis used.

This stencil calculation can be stored and applied as a SpMV operation, similar to TempestRemap. The CAAS nonlinear filter





is an online cost as it is solution dependent, and so at every instance that it is applied it carries a $\mathcal{O}(\frac{p^2}{2})$ bounds calculation cost for each of $N$ DoFs.

In terms of the computational cost, similar to TempestRemap and GMLS, the transfer for smooth function and the detection of discontinuities in WLS-ENOR can take advantage of some preprocessing steps to build matrix-based operators. Hence, they

primarily involve SpMV as the primary online cost as well. Additionally, the resolution of discontinuities requires constructing and solving the generalized Vandermonde systems for each target cell near discontinuities as described in Section 2.2, which can be relatively expensive. Hence, the overall cost for resolving discontinuities depends on the percentage of the cells in discontinuity regions, which is determined only at runtime to account for field evolution. In general, on coarse meshes, this ratio may be relatively higher. As the underlying mesh is refined, the solution field profile has a finer approximation that results

in a lower proportion of discontinuous cells. So the overall cost for the most expensive part in WLS-ENOR is expected to have a sublinear time complexity under mesh refinement.

While the theoretical complexity requirements for the remapping algorithms under consideration can be considerably different, it is imperative to measure the performance of these implementations on various architectures on standard test problems to gauge overall efficacy (accuracy vs total computational time for $N_r$ steps). This requires configuring, building, and installing

the ESMF, TempestRemap, Compadre (GMLS-CAAS), and WLS-ENOR libraries on the same machine to compare the performance profile for test problems at scale. Given the complexity and magnitude of such a task, we express this as another future avenue for research experiments that can add value to the broader climate science community using remapping algorithms.

## 7    Conclusions

Remapping algorithms are critically important in climate science applications to maintain key numerical properties during

the transformation and transfer of coupled field data between component models. Inaccurate, non-conservative, or highly dissipative remapping operators can introduce numerical artifacts in the coupled fields, which can destroy the high-order accuracy of component solvers and propagate errors into the global nonlinear system representing the climate system. Hence, understanding the behavior of remapping algorithms for earth system modeling requires standardized numerical definitions that provide better insight into the accuracy of transferred fields, along with the ability to preserve global and local solution

bounds to avoid numerically induced instabilities in the system.

In this paper, with these motivations in mind, four different remapping schemes were selected based on the current availability of the software implementations, the maturity of the underlying numerics, and potential computational efficiency that could be obtained for real-world scenarios in comparison to existing state-of-art remapping algorithms used in production climate simulations. The mesh-based implementations in the libraries ESMF (Hill et al., 2004) and TempestRemap (Ullrich

and Taylor, 2015; Ullrich et al., 2016) provide support for the projection of field data between unstructured grids by using intersection meshes to compute conservative weights through global $L^2$-minimization approaches. These remapper implementations are used routinely in several earth system solvers such as CESM (Hurrell et al., 2013) and E3SM (E3SM Project, 2018) for remapping scalar and flux field variables between component models. In contrast to the intersection-mesh-based remap-





algorithms cover a large span of low and high order conservative solution transfer implementations that can directly impact the
overall stability and accuracy of predictive solvers and analysis suites for weather and climate modeling.

Comparing these four distinct remapping algorithms requires a uniform test infrastructure, which provides the framework
to create new verification studies and analyze the metrics obtained from the remapping algorithms. To enable such unbiased
comparative studies, we introduced several remapping metrics that represent the key properties of remapping algorithms. These
include global error measures under various norms like $\|E\|_{L_1}$, $\|E\|_{L_2}$, $\|E\|_{L_\infty}$, and gradient error measures given by $\|E\|_{H_1}$
and $|E|_{H_1}$. These error norms provide the necessary verification of the theory and implementation of the remapping algorithms
by measuring the theoretical order of accuracy applied to solution fields with sufficient smoothness. Next, global conservation
errors are measured by comparing the integral of the sampled reference fields on $\Omega_t$ against the remapped solution on $\Omega_t$ for
multiple projection iterations. Finally, metrics for the global and local departure away from maxima or minima in the solution
fields can be evaluated in various norms to provide insight on monotonicity preservation and feature dissipation due to the
remapping algorithms. These standardized metrics provide the blueprint to build the remapping intercomparison suite, which
was then used to understand the numerical properties of all algorithms under consideration.

Furthermore, a flexible workflow built on several Python-based drivers to generate the unstructured meshes of different
element topologies and resolutions, including regionally refined meshes, and to accurately sample five element-averaged fields
using SPH expansions was provided. With these input meshes, the four remapping algorithms were applied for both the smooth
analytical fields and representative real fields in an iterative fashion to compute cyclic projections ($\Omega_s \rightarrow \Omega_t \rightarrow \Omega_s$) for FV-FV
field transfers between component models in a climate system.

The results compiled from various test problems demonstrate that the conservative remapping implementations in ESMF
are all first-order accurate for smooth problems, even though the 'conserve2nd' option produces consistently better accuracy
than the 'conserve' option. Hence, when possible, the ESMF 'conserve2nd' option should be used to obtain better remaps
for climate simulations. In contrast, TempestRemap produces remaps that are globally conservative and high-order accurate
up to $\mathcal{O}(h^3)$ convergence rates for smooth solution fields using $p = 2$. These low order ESMF maps are highly dissipative in
general and can be rectified even with $\mathcal{O}(h^2)$ maps produced with TempestRemap, which show bounded dissipation even for
sharp features. However, neither one of these low and high-order $L^2$-minimization approaches can guarantee monotone remaps
without the use of post-processing filters such as slope limiters or CAAS to enforce global and local solution bounds. The use
of these filters, however, inevitably introduces additional dissipation that can become significant with repeated applications of
the remap operator.

On the other hand, the hybrid mesh-based and meshless schemes achieve very high order consistently (up to $\mathcal{O}(h^5)$) while
retaining global conservation, which has been one of the key advantages of traditional overlap-mesh-based schemes imple-
mented in ESMF and TempestRemap. Additionally, the ability to capture smoothly varying fields very accurately without any





degradation even for high polynomial degree reconstructions makes these schemes attractive and competitive, as compared to traditional mesh-based schemes utilizing $L^2$-minimization methods for ESMs that require computation of overlap meshes, which can incur a significant computational cost at high spatial resolutions (Mahadevan et al., 2020). These methods can especially be valuable when highly accurate scalar fields need to be sampled on a refined RLL grid for further analysis or in-situ

visualization to track multidecadal climate evolution.

It is essential to note that while the fully mesh-based remapping algorithms are, in general, insensitive to mesh resolutions or the topology, both the hybrid and meshless schemes are liable to larger mesh-dependent dissipation. This is especially evident when the source and target mesh resolutions differ drastically. The use of the CAAS algorithm, when combined with even non-conservative schemes such as GMLS, can provide global conservation and bounds preservation at the cost of added

dispersion to control numerical oscillations. In contrast, the built-in discontinuity indicators used by the WLS-ENOR algorithm demonstrate good feature resolving properties for real fields in all experiments conducted.

These experiments conducted on both uniform resolution meshes and regionally refined meshes provide valuable insight into the properties of remapping algorithms and their numerical behavior. However, practical use of these algorithms in real-world scenarios requires deeper investigation using more topologically diverse, complex meshes and discretization specifications that

are more representative of components used in E3SM and CESM.

Finally, we want to emphasize that the MIRA infrastructure presented in this paper is freely available as an open-source package (Guerra et al., 2021) to compare new and existing remapping algorithms under the same overall test constraints. Such intercomparison studies are important to evaluate the cost of remapping algorithms under stability and accuracy constraints, which remain crucial to better understanding the propagation of errors in coupled climate and weather systems.

*Code availability.* Information on the availability of source code for the remapping metrics intercomparison infrastructure featured in this paper, all relevant input meshes, and the final consolidated metrics data for schemes are provided below.



| Artifact | Availability, License, Links |
|----------|------------------------------|
| **Remapping intercomparison code** | The Python intercomparison workflow infrastructure and the scripts to compute the metrics data for remapping schemes are available (Guerra et al., 2021). This code was developed through funding from the CANGA project, and is publicly released under an open-source license, with copyright owned by UChicago Argonne, LLC. v1.0 was tagged in September 2021 and the archive is available for download from Zenodo. The original repository is hosted in GitHub. DOI: 10.5281/zenodo.5518037. |
| **Raw satellite data snapshot** | The raw satellite data snapshots used to generate the Spherical Harmonic decomposition for TPW, CFR, and TOPO fields are available at (Guerra and Mahadevan, 2021). The files are distributed under the Creative Commons Attribution 4.0 International License, while acknowledging the original sources for the data for TPW (Wimmers and Velden, 2011), CFR (Platnick et al., 2020), and TOPO (Center, 2009; Amante and Eakins, 2009). v1.0 was released on Aug 09, 2021 and available here. DOI: 10.5281/zenodo.5172792. |
| **Input meshes and output metrics data** | The pre-processed input meshes that were used in the study are available at (Mahadevan et al., 2021). The input meshes and the consolidated metrics data for each of the remapping methods (ESMF, TempestRemap, GMLS, GMLS-CAAS and WLS-ENOR) are made available under an open-source license, with copyright owned by UChicago Argonne, LLC. v1.0 was tagged in September 2021 and the archive is available for download from Zenodo. The original data repository is hosted at GitHub. DOI: 10.5281/zenodo.5518065. |

*Author contributions.* VM, JG, XJ and PK wrote the paper (with several helpful review comments from PB, PU and RJ). JG wrote significant portions of the remapping intercomparison code (Guerra et al., 2021), with contributions from VM, and PK. VM generated the remap output data using ESMF and TempestRemap libraries. PK generated results using GMLS, and GMLS-CAAS. XJ and YL generated results using WLS-ENOR algorithms. VM consolidated the final metrics data from the remapping teams, performed analysis of the results, and generated

5   the necessary data for the algorithmic intercomparison study. The broader project idea was conceived by PJ and PU.

*Competing interests.* The authors declare that they have no conflict of interest.

*Disclaimer.* Any subjective views or opinions that might be expressed in the paper do not necessarily represent the views of the U.S. Department of Energy or the United States Government.



*Acknowledgements.* This material is based upon work supported by the U.S. Department of Energy, Office of Science, Office of Advanced Scientific Computing Research and Office of Biological and Environmental Research, Scientific Discovery through Advanced Computing (SciDAC) program. Additional support was provided by NOAA Office of Oceanic and Atmospheric Research under the NOAA-University of Oklahoma cooperative agreement (NA11OAR4320072), US Department of Commerce. We gratefully acknowledge the computing resources provided on Bebop, a high-performance computing cluster operated by the Laboratory Computing Resource Center at Argonne National Laboratory, which is supported by the Office of Science of the U.S. Department of Energy under contract DE-AC02-06CH11357, to generate the output for test problems using ESMF and TempestRemap remapping software libraries. We also gratefully acknowledge use of the Common Engineering Environment (CEE) Compute Servers at Sandia National Laboratories to generate the output for test problems using GMLS. Computational results for WLS-ENOR were obtained using the Seawulf cluster at the Institute for Advanced Computational Science of Stony Brook University, which was partially funded by the Empire State Development grant NYS #28451.



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
