# Peer review of "Metrics for Intercomparison of Remapping Algorithms (MIRA) applied to Earth System Models"

_Geoscientific Model Development, 2021_

## Referee Comment (RC1)

**Review of Metrics for Intercomparison of Remapping Algorithms (MIRA) applied to Earth System Models**

by Mahadevan et al, January 2022

This paper presents a methodology for evaluating regridding algorithms and applies it to compare four different libraries: ESMF Regrid, TempestRemap, Generalized Moving-Least-Squares (GMLS), and WLS-ENOR. The methodology proposes a set of metrics to measure the quality of the regridding based on different criteria: sensitivity, consistency, conservation, monotonicity, dissipation.

The paper starts with an in-depth review of the mathematical basis of current regridding algorithms used in climate modelling. Then it defines metrics to evaluate the criteria listed above and presents the workflow to calculate the metrics for various meshes at different resolutions for different fields. Three quasi-uniform (Cubed-Sphere, Voronoi, and Regular Latitude-Longitude) meshes at five different resolutions and two regionally refined (Cubed-Sphere RRM, Voronoi RRM) meshes at three different resolutions are used. The regridding of two analytical fields and three real fields (Total Precipitable Water, Cloud Fraction, Global Topography) are studied.

To analyse the regridding consistency, the convergence rates with respect to the mesh refinement for quasi-uniform grids are calculated for the different libraries. The 2nd order regridding for ESMF Regrid and TempestRemap show degraded convergence rates in some cases while GMLS (and GMLS-CAAS i.e. applying a post-processing filter) and WLS-ENOR (that do not require computation of mesh intersection) show high-order consistency. Regarding global conservation, ESMF Regrid and TempestRemap are conservative by construction while GMLS is not; however, GMLS-CAAS and WLS-ENOR show good global conservation. The meshless and hybrid remap schemes, GMLS-CAAS and WLS-ENOR, preserve the global extrema conservation (monotonicity) while mesh-based algorithms implemented in ESMF Regrid and TempestRemap do not. The paper also provides a detailed analysis of the local extrema preservation for the four libraries and on the differences brought by using regionally-refined meshes.

The paper is well written and structured. As such, it provides an interesting contribution to the theory and practical implementation of code coupling for Earth System Modeling even if the precise impact of the conclusions for real-world coupled systems are not straight forward to infer. Overall, I recommend that the paper be accepted for publication after taking into account the comments I list here below.

**Important comments**

- 1. The notion of grid resolutions and convergence rate (as a function of the grid refinement) are central to the paper but are not well defined:
  - p.16, L.11 : in « uniform mesh refinement », are you talking about source or target mesh, or both ? Or do you include the "cross-resolutions", e.g. the calculations done for CS(0) MPAS(4)?

- p.24, L.6: what are exactly the 5 (uniform) and 3 (refined) resolutions used for the calculations? Please describe them more precisely as this is important for the calculation of the convergence rate.
- p.25, L.8: How you calculate the convergence rate (as a function of *h* the mesh refinement)? Please give more detail, this is central to the paper.
- p.25, L.9: Please better justify (or refer precisely to numbers in Table 1) why you state "ESMF first-order conservative scheme yields expected rates"? What rates do you expect? At the bottom of p.24, you provide some details on the expected convergence rates but the definition involves *h* and I don't think that *h* has been defined precisely although it is clearly linked to the grid resolution (see my comment above).
- p.29, L.1: again, please detail what are the "theoretically expected rates"; how do you evaluate them? Please illustrate why you state that "theoretically expected rates are observed" by referring precisely to numbers on Table 4.
- p.37, L.4 and p.38, Table 6. Be more precise on the resolution used for each grid; what does "the finest CS-MPAS RRM mesh combination" mean precisely?
- Figure 7-8-9-10-11: Define more precisely CS(0), CS(4), MPAS(0), MPAS(4) (see also my comment above for p.24, L.6.
- p.39, Fig. 13: Define more precisely CS-RRM(0), MPAS-RRM(0), CS-RRM(2), MPAS-RRM(2). In the caption, put "for a) coarse to fine, and b) similar refinements ..."
- 2. For all figures, the x and y axes should be redrawn with bigger and clearer fonts.
- 3. Figure 6:
  - I don't understand the y axis. How can you have negative values for those error norms? Given equations (7) and (9), I don't think this is possible.
  - I suppose that each curve is for one specific pair of grids a specific resolution. For example, the left-most plot is for the grid pair CS-MPAS with a specific resolution of CS (among the 5 possible) and a specific resolution of MPAS (among the 5 possible). If I am right, please indicate which is the resolution for CS and which is it for MPAS.
- 4. p.32, L8-9 and Figures 8 and 9:
  - I think Lmin should be Gmin .and Lmax should be Gmax and refer to equations (12) and (13) as you are describing here global extrema and not local extrema.
  - Fig. 9 a): How can Gmin be negative (for TempestRemap)?
  - Fig 9 b): Please specify in the captions where is the TempestRemap curve?
- 5. p.37, L.5: You write "The global errors with respect to all error norms are considerably smaller in WLS-ENOR and TempestRemap." This is particularly true for analytical function but not so clear for real fields. Can you comment on this on the text?
- 6. p.42, L.25: you completely exclude here "dynamic" grids, i.e., grids which definition evolve with time and for which the regridding weigths have to be recalculated at each timestep. No "offline" operations for those grids. Please comment on this.

Other comments:

• p.3, L.8-9 : This is not true for YAC. YC is a full coupler, it is not an interpolation library designed to generate weights to be consumed by another coupler.

- p.9, L5-7 : I think that SCRIP 1st order conservative remapping is indeed similar to ESMF 1st order conservative remapping, but I don't think this is true for the 2nd order. Compared to the 1st order, SCRIP applies weighted gradients in the longitudinal/latitudinal direction, while ESMF applies Kritsikis 2017, so SCRIP and ESMF are different for the 2nd order.
- p.9, L20-21 : I guess ESMF patch algorithm would also fall in this catergory ?
- p.10, L.22 : I understand that the polynomials are integrated over each mesh of the supermesh, but then can you give some details on the procedure to go from the supermesh to the target mesh ?
- p.16, L.8 : can you detail why you write that L1 identifies errors in large-scale features and that L2 identifies errors in small-scale features ? I never understood that.
- p.20,Fig.2 : there should be an arrow between the source and target mesh definition (top right) and the bottom left of the figure, which represents the regridding per se as the mesh definition is certainly required for the regridding step
- p.24,L.13 : I think that if (Nunitype) is the number of uniform grids, N sould be defined as

$$N = [(N^{uni}_{type})! / 2] * [N^{uni}_{ref}]^2 * [N_{fields}]$$

and

 $N = [(N^{rrm}_{type})! / 2] * [N^{rrm}_{ref]}]^2 * [N_{fields}]$

It happens here that N, as expressed in the paper, i.e N =  $[(N^{uni}_{type})! / 2] = 3$  for  $(N^{uni}_{type})=3$  and that N =  $[(N^{rrm}_{type})! / 2] = 1$  for  $(N^{rrm}_{type})=2$ , gives the right result but this is just by chance.

- p.24,L.30 : I had problem remembering what *p* is. I understand that it is, e.g. for TempestRemap the order of the polynomials used for the reconstruction? It would probably be useful to explicitly note this "definition" of *p* in 2.3.2. and in 2.3.4. (For GMLS, p is clearly defined on p.11 L.28, which is good.)
- p.32, L.3: "for some scalar fields": please provide example of those intensive variables such as SST
- p.32, L.12: recall which are the "mesh-based remapping schemes", i.e., ESMF and TempestRemap.
- p.33, L.3: Please clarify in the text and in the captions where is TempestRemap on Fig. 10 c)
- p.37, Fig.12: Specify the metric in the captions
- p.37, L.15-17 compared to L.18-19: for me, those two sentences are contradictory: if 'conserve2nd' is only marginally better than 'conserve', how can you state that the superiority of the 'conserve2nd' is clearly demonstrated?
- p.42, L.3: When discussing extension to vector fields, you should also mention that regridding the vector components expressed in a local coordinate system linked to the grid or in the spherical reference system is wrong in principle. For proper treatment of vector fields, the source code should send the 3 components of the vector projected in a Cartesian coordinate system as separate fields. The target code should receive the 3 interpolated Cartesian components, recombine them to get a proper vector field, and project the resulting vector in its local reference system.

Minor comments:

- p.1, L.3 : I think the form 's can be used for people only. Here « component » is only a qualifier and I think you should write « one component computational mesh »
- p.3, L.30 : « interpolators »  $\rightarrow$  « interpolations »?
- p.5, L.6 : « interpolator »  $\rightarrow$  « interpolation »?
- p.6, L31 : I suppose that both « discontinuity detecting » and « a posteriori stabilization » both qualify the « procedure » ? If so I would write « ... discontinuity-detecting and a-posteriori-stabilizing procedure »
- p.9, L.28 : I would change « weather and climate modeling » for « weather and climate applications»
- p.10, L.15-16 : put « (FV) » after « finite volume »
- p.10, L29 : for « potential function », do you mean the potential function expressing the tractory of the mesh boundary ?
- p.14, L.21 : repeated **back-and-forth** remap transfers ?
- p.14, L.24 : I think you should remove the "psi" after « the regridded field » as the regridded field is "R Ds psi"
- p.29, L12-14:, I don't think it because of the presence of discontinuities that you do a remap comparison? Please rephrase.
- p.45, L.27: I would put part of the sentence, i.e., "These low order ESMF maps are highly dissipative" with a "However," before the sentence starting with "In contrast, ..." that introduces TempestRemap and then go on with the remark on the fact that the highly dissipative ESMF maps that can be corrected by the O(h2) TempestRemap maps.
- p.45, L.27: I would also give a general definition of *p* and *h* to make sure that someone reading only the introduction and the conclusion can understand the main findings of the paper.

---

## Author Comment (AC2)

**Author comment for Anonymous Referee #1**

Vijay S. Mahadevan and Jorge E. Guerra on behalf of all authors

February 17, 2022

Dear Reviewer,

We sincerely appreciate the detailed comments and suggestions provided to improve the paper. We will address all of the major comments and fix the minor issues pointed out. To provide some context and to answer some relevant questions, please look at the selected inline discussions below.

1. The notion of grid resolutions and convergence rate (as a function of the grid refinement) are central to the paper but are not well defined

Thanks for the comment. We understand that some of the details have not been explicitly provided in the manuscript, but rather linked through an external repository[1] containing the raw mesh files used in the study. However, we will include all the relevant details in the text to avoid any further confusion.

2. p.16, L.11 : in "uniform mesh refinement", are you talking about source or target mesh, or both ? Or do you include the "cross-resolutions", e.g. the calculations done for CS(0) - MPAS(4)?

In the context of uniform mesh refinement, yes we are referring to uniform resolution increase for both the source and target meshes. The spatial error component is generally a function of $C_s\mathcal{O}(\mathcal{E}_s)+C_t\mathcal{O}(\mathcal{E}_t)$, where $\mathcal{E}_s, \mathcal{E}_t$ are the chosen error norms computed on source and target meshes and $C_s, C_t$ are some constants respectively. For the convergence estimation, we utilize similar resolutions on both meshes (in terms of average element jacobian size) e.g., error metrics on $CS(i) - MPAS(i), \forall i \in [0, 4]$ were chosen for convergence rate calculations. The cross-resolutions can be utilized as well, but generally the convergence rate can plateau when the
* * *
[1]MIRA Datasets: https://github.com/CANGA/MIRA-Datasets

sampling error in either $\Omega_{h,s}$ or $\Omega_{h,t}$ starts dominating the remap errors (especially with coarse resolutions), leading to inaccurate convergence rate estimates.

3.  p.24, L.6: what are exactly the 5 (uniform) and 3 (refined) resolutions used for the calculations? Please describe them more precisely as this is important for the calculation of the convergence rate.

A better description of the uniform and refined mesh resolutions will be provided. Relevant links to the mesh files in our open-source repository will be added here again.

4.  p.25, L.8: How you calculate the convergence rate (as a function of h the mesh refinement)? Please give more detail, this is central to the paper.

To compute the accuracy convergence rate estimates, we utilize the average element size for each mesh (especially for the quasi-uniformly refined meshes). The slope of the curve plotted between the various error norms and the average element size in the log-log scale gives the numerical convergence rate for the method, for that combination of meshes. We will provide more context and explain this in better detail.

5.  p.25, L.9: Please better justify (or refer precisely to numbers in Table 1) why you state "ESMF first-order conservative scheme yields expected rates"? What rates do you expect?

ESMF implements both the first and second order remapping methods. The statement specifically refers to the first order method. We will clarify that better since the second order method implementation in ESMF (v8.1) still only yields first order convergent method.

6.  At the bottom of p.24, you provide some details on the expected convergence rates but the definition involves h and I don't think that h has been defined precisely although it is clearly linked to the grid resolution (see my comment above).

This will be described better when we make changes to the manuscript.

7.  p.29, L.1: again, please detail what are the "theoretically expected rates"; how do you evaluate them? Please illustrate why you state that "theoretically expected rates are observed" by referring precisely to numbers on Table 4.

Typically, using the global error norm metrics, the theoretically expected rates are $O(h^{p+1})$, where $p$ is the order of polynomial reconstruction. Table 4 provides the convergence rates for WLS-ENOR schemes with different polynomial orders. For $p = 2$ and $p = 4$, the rates follow the definition of $O(h^{p+1})$, while we observe some superconvergence with $p = 3$ due to some error cancellations. We can expand the text to describe the behavior.

8.     p.37, L.4 and p.38, Table 6. Be more precise on the resolution used for each grid; what does "the finest CS-MPAS RRM mesh combination" mean precisely?

Figure 7-8-9-10-11: Define more precisely CS(0), CS(4), MPAS(0), MPAS(4) (see also my comment above for p.24, L.6.

We will provide the exact resolutions used for these cases.

9.     p.39, Fig. 13: Define more precisely CS-RRM(0), MPAS-RRM(0), CS-RRM(2), MPAS-RRM(2). In the caption, put "for a) coarse to fine, and b) similar refinements

Thanks for pointing this out. We will define it more precisely.

10.     For all figures, the x and y axes should be redrawn with bigger and clearer fonts.

The figures will be regenerated and updated during manuscript revision.

11.     Figure 6: I don't understand the y axis. How can you have negative values for those error norms? Given equations (7) and (9), I don't think this is possible.

Figure 6 is a semi-logy plot, where the error norms in Y-axis are essentially log(error). Equation (7) and Equation (9) only provide the error metric. We will make the Y-axis title clearly state this.

12.     Figure 6: I suppose that each curve is for one specific pair of grids a specific resolution. For example, the left-most plot is for the grid pair CS-MPAS with a specific resolution of CS (among the 5 possible) and a specific resolution of MPAS (among the 5 possible). If I am right, please indicate which is the resolution for CS and which is it for MPAS.

That is correct. We chose the finest resolution pair in each grid combination for that study. Text will be updated to reflect the information.

13. I think Lmin should be Gmin and Lmax should be Gmax and refer to equations (12) and (13) as you are describing here global extrema and not local extrema.

This is correct. We will fix it.

14. Fig. 9 a): How can Gmin be negative (for TempestRemap)?

Yes, Gmin cannot be negative by definition. We will re-verify the results again for this metric and ensure the right data is getting plotted.

15. Fig 9 b): Please specify in the captions where is the TempestRemap curve?

The TempestRemap curve is at the zero-axis. I can see why the figure is misleading. We will explain the observed results better in the text and perhaps modify the scale to show all the lines more clearly if possible.

16. p.37, L.5: You write "The global errors with respect to all error norms are considerably smaller in WLS-ENOR and TempestRemap." This is particularly true for analytical function but not so clear for real fields. Can you comment on this on the text?

Yes we will update the text to better clarify the conclusions.

17. p.42, L.25: you completely exclude here "dynamic" grids, i.e., grids which definition evolve with time and for which the regridding weigths have to be recalculated at each timestep. No "offline" operations for those grids. Please comment on this.

For dynamic grids, the remapping cost is fully incurred at runtime. Note that while computing the linear map on the global domain for every small change could be the trivially simpler (and more expensive) option, including feature-tracking algorithms can help offset a majority of the cost by localizing recomputation work to only the adapted regions. We will add a short discussion about dynamic grids in the performance section.

---

## Author Comment (AC3)

**Author comment for Anonymous Referee #2**

Vijay S. Mahadevan and Jorge E. Guerra on behalf of all authors

February 20, 2022

Dear Reviewer,

We are grateful for the suggestions and technical comments provided to improve the manuscript. We will address all the technical and non-technical comments when revising the manuscript. Additionally, we have provided a discussion on some of the general comments in the reviewer notes, which should provide better context to the work presented in the paper.

1. I suggest a brief review of and possible reference at "https://cerfacs.fr/wp-content/uploads/2021/11/Globc_TR_Valcke_21_145_regridding_analysis_final.pdf"

   Yes, we will include a reference to the technical report, which certainly looks more indepth at the performance aspects of some of the remapper implementations.

2. Please define convergence and convergence rate somewhere.

   We will include these definitions in the consistency Section 3.2 of the manuscript.

3. Please explain the numerical results in Section 5 in slightly greater detail. I understand you have defined the metrics, but I believe there needs to be some more detailed description of what the values represent qualitatively to help the reader fully understand the tables and figures, especially tables associated with convergence rates when first shown in Section 5.1.1.

   Understood. Reviewer #1 had a related comment as well. We will address both of these comments by adding more details about the theoretically expected and the numerically observed convergence rates behaviors. The additional details should add more context quantitatively understand the results showcased in the tables.

4. You allude to multiple resolutions on page 24 and you show results for varied resolution in the paper including coarse to fine and reverse. But I don't think you ever define the resolutions. Could you provide some insight into the actual resolutions associated with each grid and refinement. Maybe a table? Number of gridpoints would be fine.

We can certainly add a table with some statistics about each of the meshes including the resolution (number of elements), average/min/max element sizes to provide better context for both uniform and regionally refined meshes. Just a note that the meshes are available openly in our data repository[1] as well.

5. As I understand it, the methods assessed are all linear and are implemented as a sparse matrix multiply applied to the source data, except for the special extra methods (ie. CAAS) that are implemented as run-time adjustments the require analysis of the actual data fields.

This is true. The methods that we have currently explored are either linear (ESMF, TempestRemap) or quasi-linear (GMLS-CAAS, WLS-ENOR), where the bulk of the work can still be performed once with a bounded cost at runtime to apply the projection for field transfer.

6. Is there anything to be said about non-linear methods? Could ESMs benefit? Could they be easily implemented? I don't think this paper needs to address this question.

Nonlinear remap methods in general have a much higher computational complexity in comparison to linear or quasi-linear remapping schemes. However, such methods can offer tremendous flexibility in terms of imposing conservation requirements, preserving inherent properties of the field with additional constraints in the system (e.g., divergence-free conditions), maintaining valid bounds of the transferred field data (monotonicity), producing optimally accurate approximations (discontinuity detecting, feature tracking, adaptive-order reconstructions) to provide coupled data with minimal spatial error propagation. Additionally, since the nonlinear methods do not need to rely on explicit mesh connectivity through creation of a linear map, the application of such schemes to adaptive or moving meshes are a trivial extension. So there is certainly a lot of value in having a nonlinear remap method tuned for ESMs, if there are specific components like sea-ice that require high-order accurate field data
* * *
[1] MIRA Datasets: https://github.com/CANGA/MIRA-Datasets

satisfying auxiliary conditions.

We have provided some references related to these schemes in Section 2.1 and Section 6.3 in the manuscript, which also provide significantly more information for interested readers.

---

## Author Response (AR1)

**Author comments explaining change-diffs in manuscript**

Vijay S. Mahadevan on behalf of all authors

June 4, 2022

Dear Editor,

We are grateful to the two anonymous reviewers for useful suggestions and comments to improve the manuscript. We have addressed all the comments when revising the manuscript. The pdf of the differences generated with *latexdiff* has been attached and the changes are explained below.

**Explanation of changes in *latexdiff* file**

1. Author name addition: Dr. David Marsico, as mentioned in author comment.

2. P.1 L.3-4: Better rephrasing to make abstract clearer based on RC1.

3. P.1 L.19: Remove brackets for acronyms to improve readability.

4. P.2 L.19: Formatting change.

5. P.3 L.9: Remove YAC interpolator reference as suggested in RC1.

6. P.3 L.12-15: Provide context and add reference to the regridding analysis work by Sophie Valcke, as suggested in RC2.

7. P.3 L.35: Clarity (RC1).

8. P.5 L.12-13: Clarity (RC1).

9. P.7 L.4: As suggested in RC1.

10. P.9 L.25: To address question about patch remap in ESMF (RC1).

11. P.10 L.4: As suggested in RC1.

12. P.10 L.20: Addressing comment about better defining degree $p$ (RC1).

13. P.10 L.22: As suggested in RC1.

14. P.11 L.2: References to further details about overlay based remap as requested in RC1.

15. P.11 L.3-5: Addressing comment in RC1 about potential functions in TempestRemap.

16. P.13 L.8: Addressing comment about better defining degree $p$ (RC1).

17. P.14 L.31: As suggested in RC1.

18. P.15 L.3: As suggested in RC1. Removed $\psi$.

19. P.15 L.24-26: Clarifying question raised in RC1 about how norms help understand errors in large-scale and small-scale features.

20. P.16 L.2-9: Addressing comments from RC1 and RC2 to better explain the idea behind convergence order measurements and what is expected theoretically for a given degree $p$.

21. P.18 Equation (12): Addressing comment by RC1 about the Global minima metric, which was wrongly defined (sign changed). Text (P.18 L.5-8) has been added to clarify the changes further.

22. P.19 L.9-15: Addressing comment in RC2 about the validity of repeated remap study in production climate simulations.

23. P.24 L.7 - P.25 L.3: Addressing comments in RC1 and RC2 about more details on the meshes used along with information on the number of elements and nodes in each case. The newly added Table 1 also provides these details explicitly.

24. P.25 L.10-11: Correcting definition of $N$. We implicitly assumed that the combination of two meshes are needed for remap (one for source and one for target). This assumption is now explicitly shown as $N_{type}^{uni}C_2$ and $N_{type}^{rrm}C_2$ for uniform and regionally refined cases respectively.

25. P.26 L.11-14: More details added about how $h$ and $p$ are used in convergence study, as requested in RC1.

26. Formatting changes for 'conserve' to *conserve* and 'conserve2nd' to *conserve2nd* everywhere in the manuscript to make the text more readable.

27. P.27 L.1-2: Addressing comment in RC2 about what convergence rates of $1.00x$ and $0.99x$ imply. They both are considered $\mathcal{O}(h)$ asymptotically.

28. P.27 L.6: More description about figures added to text (RC2).

29. P.27 L.10-12: Addressing comments in RC1 and RC2.

30. P.28: Figure 6 caption improved with more details about the plot, as requested in RC1.

31. All figures have been regenerated with better fonts for axes and clearer background.

32. P.31 L.2-5: Clarifying the meaning of convergence rates, especially in relation to degree $p$ and mesh size $h$ as requested in RC1.

33. P.33 L.5-10: Addressing questions related to why TempestRemap accumulates errors more quickly than ESMF (RC2).

34. P.34 L.7: Including suggestion in RC1.

35. P.34 L.13: Replace $L_{max}$ with $G_{max}$ as correctly suggested in RC1.

36. P.34 L.14: Replace $L_{min}$ with $G_{min}$ as correctly suggested in RC1.

37. P.34 L.17: As suggested in RC1 to include the names of mesh-based schemes.

38. P.34 L.22-24: Addressing question in RC2 on details related to the dampening properties of ESMF maps. This cannot be fully understood without a full spectral analysis, and we do not currently have the time to pursue this.

39. P.34 L.30-31: Rephrase as requested in RC1.

40. P.35 L.3-5: Address comments about hidden traces in plots (RC1). This is now explained well in the text to avoid confusion.

41. P. 36, Figure (9): Fully regenerated as the $Gmin$ metric computed for TempestRemap and ESMF were using an older definition. This is now fixed. All $Gmin$ values should be zero or strictly negative (for deviation away from monotone solution).

42. P.37 L.4-5: Address more comments about hidden traces in plots (RC1).

43. P.39 L.8-33: Address multiple comments in RC1 and RC2 related to drawing better conclusions from the data presented in Table 7. Our changes should now explain more clearly the impact of using high-order vs low-order methods for both smooth and discontinuous field remaps.

44. P.40 L.2-11: Address comments in RC1 about replacing $L_{max}$ with $G_{max}$, hidden traces in the plots, and better explanations about behavior of high-order TempestRemap maps for preserving global bounds.

45. P.41 Figures 12-14: Update caption to provide more context (RC1).

46. P.44 L.31-33: Include comment in RC1 about care needed for vector field remaps.

47. P.45 L.18-21: Add description of dynamic meshes and how performance complexity can be calculated for such cases (RC1).

48. P.47 L.9: Add reference to Valcke et al., (2022) as suggested in RC2.

49. P.48 L.16-23: Provide more context about $h$ and $p$ (RC1). Also more details about overall conclusions derived from the data presented in the manuscript (RC1, RC2).